# Towards Bounding Causal Effects under Markov Equivalence

**Alexis Bellot**

Google DeepMind, London, UK

## Abstract

Predicting the effect of unseen interventions is a fundamental research question across the data sciences. It is well established that in general such questions cannot be answered definitively from observational data. This realization has fuelled a growing literature introducing various identifying assumptions, for example in the form of a causal diagram among relevant variables. In practice, this paradigm is still too rigid for many practical applications as it is generally not possible to confidently delineate the true causal diagram. In this paper, we consider the derivation of bounds on causal effects given only observational data. We propose to take as input a less informative structure known as a Partial Ancestral Graph, which represents a Markov equivalence class of causal diagrams and is learnable from data. In this more "data-driven" setting, we provide a systematic algorithm to derive bounds on causal effects that exploit the invariant properties of the equivalence class, and that can be computed analytically. We demonstrate our method with synthetic and real data examples.

## 1 INTRODUCTION

Causal relations are a prominent paradigm to describe our interactions with the world around us. We rely on them to make sense of notions of fairness, extrapolation, and safety in AI systems that play an increasingly important role in society. At the center of the notion of causality lies the idea of manipulation or intervention. A typical question in this context could be: "What would happen to outcome $Y$ if $X$ were set to $x$?". For example, a physician might be interested in how a biomarker $Y$ responds to a new dosage $x$ of drug $X$; or, an economist might wonder about the trajectory of economic indicators $Y$ under an interest rate hike $X = x$

in a given country $Z = z$. These questions all appeal to the same formal machinery, they aim at establishing facts about (conditional) *causal effects*, *e.g.*, written $P_x(y \mid z)$.

In general, it is impossible to infer the effect of interventions from data alone (without physically manipulating reality) as further domain knowledge or assumptions are typically needed to uniquely pin down causal effects. This motivates the study of a problem known as *(partial) causal identification* (Pearl, 2009). The idea is to combine data from an observational distribution $P(\boldsymbol{V})$ with partial knowledge of the domain, articulated as a causal diagram, to bound a causal effect $P_x(y \mid z)$ within a tight interval. In other words, the problem is to infer a set of values that contains all effects implied by the causal models consistent with the data and assumptions. If the effect can be uniquely determined it is said to be point identified and the interval reduces to a single value.

One of the foundational results in the literature is due to Manski (1990); Robins (1989). The authors showed that causal effects could be bounded with observational data without making any assumptions on the structure or causal diagram of the underlying data generating mechanisms. This approach has since been generalized to bound causal effects under instrumental variable assumptions (Robins, 1989), and given more general causal diagrams (Zhang and Bareinboim, 2019; Zhang, 2020). In parallel, several authors have shown that with a sufficiently expressive parameterization of the underlying causal model, bounds can also be computed by making inference on model parameters, with recent proposals developing polynomial optimization programs (Balke and Pearl, 1997; Hu et al., 2021; Padh et al., 2022; Li and Pearl, 2022) and Bayesian methods (Chickering and Pearl, 1996; Zhang et al., 2021; Finkelstein and Shpitser, 2020). Many of these recent works develop bounds under various assumptions about the structure and form of the underlying data generating mechanism. In practice, this formulation is often found too rigid for many practical applications as assumptions are hard to justify and test, sometimes even known to be unrealistic. A sensible concern is that forcing

*Accepted for the 40th Conference on Uncertainty in Artificial Intelligence* (UAI 2024).

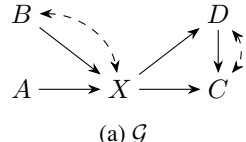

(a) $\mathcal{G}$

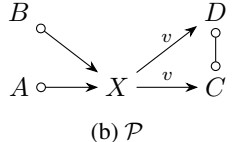

(b) $\mathcal{P}$

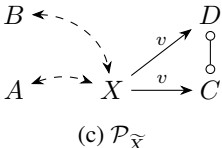

(c) $\mathcal{P}_{\widetilde{X}}$

Figure 1: Examples of diagrams.

a single diagram or parametric model family may lead to false modeling assumptions and misleading inferences on causal effects.

In the spirit of designing more "data-driven" AI systems, one approach to circumvent the need for prior knowledge is to learn the causal diagram from data first, and then perform identification from there. The statistical constrains found in data (e.g. conditional independencies) can be leveraged to infer a class of Markov equivalent (ME) causal diagrams that is commonly represented as a Partial Ancestral Graph (PAG) (Richardson and Spirtes, 2002; Spirtes et al., 2000; Zhang, 2008b). For example, the PAG $\mathcal{P}$ in Fig. 1b encodes the ME class of causal diagrams that would be consistent with data generated according to the causal diagram $\mathcal{G}$ in Fig. 1a. In the PAG $\mathcal{P}$ the directed edges encode ancestral relations, not necessarily direct, and the circle marks stand for structural uncertainty. Directed edges labeled with $v$ signify the absence of unmeasured confounders. Identification (determining whether causal effects may be uniquely computed) from PAGs is of increasing interest. Several recent techniques for the identification of causal effects have been developed (Zhang, 2008a,b; Jaber et al., 2018a; Hyttinen et al., 2015; Perkovic et al., 2018; Jaber et al., 2019b, 2018b) including a calculus and complete algorithms (Jaber et al., 2019a, 2022).

In this paper, we pursue a generalization of the partial identification task that consists of bounding causal effects with more restricted domain knowledge in the form of a class of ME causal diagrams (instead of a fully specified causal diagram). This notion is "data driven" in the sense that equivalence classes can, in principle, be inferred from observational data $P(\boldsymbol{V})$ only, up to an assumption of faithfulness[1]. Our main contributions is to show that the data entails constraints on the value of causal effects that can be exploited to derive tighter bounds than previously considered. More specifically, we summarize our contributions as follows.

- **Section 4**. We derive analytical expressions (closed-form, in terms of $P(\boldsymbol{V})$) for lower and upper bounds on a causal effect of interest given observational data based on the structure of a PAG (Alg. 1). In particular, we show that the proposed bounds outperform the bounds due to Manski

(1990); Robins (1989) in general (Prop. 5) and provide several examples.

- **Section 5**. We investigate enumeration strategies, *i.e.* the strategy of listing ME causal diagrams and performing partial identification on each diagram separately using existing "diagram-specific" bounding techniques. We show that, in fact, a large portion of ME causal diagrams could be shown to be "redundant" for the purpose of bounding causal effects (Props. 7 and 8). Despite this simplification, still, we conjecture that a large number of graphs (increasing with the number of nodes) must be considered in general (Prop. 9), which suggests that enumeration strategies might be computationally intractable.

## 1.1 PRELIMINARIES

We use capital and small letters to denote random variables and their values respectively, *e.g.* $X$ and $x$, and bold capital and small letters to denote sets of variables and their values, *e.g.* $\boldsymbol{X}$ and $\boldsymbol{x}$. We use $P(\boldsymbol{x})$ as an abbreviation for probability $P(\boldsymbol{X} = \boldsymbol{x})$, and similarly for conditional probabilities. For sets of variables $\boldsymbol{X}, \boldsymbol{Y}, \boldsymbol{Z}$, conditional independence in $P$ is denoted $(\boldsymbol{X} \perp\!\!\!\perp \boldsymbol{Y} \mid \boldsymbol{Z})_P$ and d-separation[2] in a graph $\mathcal{G}$ is denoted $(\boldsymbol{X} \perp\!\!\!\perp \boldsymbol{Y} \mid \boldsymbol{Z})_{\mathcal{G}}$.

The framework that underpins causal effects and diagrams rests on Structural Causal Models (SCMs) following (Pearl, 2009, Def. 7.1.1). A SCM $\mathcal{M}$ is a tuple $\langle \boldsymbol{V}, \boldsymbol{U}, \mathcal{F}, P(\boldsymbol{U}) \rangle$, where $\boldsymbol{V}$ is a set of endogenous (observed) variables, $\boldsymbol{U}$ is a set of exogenous latent variables, and $\mathcal{F} = \{f_V\}_{V \in \boldsymbol{V}}$ is a set of functions such that $f_V$ determines values of $V$ taking as argument variables $\boldsymbol{Pa}_V \subseteq \boldsymbol{V}$ and $\boldsymbol{U}_V \subseteq \boldsymbol{U}$, i.e. $V \leftarrow f_V(\boldsymbol{Pa}_V, \boldsymbol{U}_V)$. Values of $\boldsymbol{U}$ are drawn from an exogenous distribution $P(\boldsymbol{u})$. We assume the model to be recursive, i.e. that there are no cyclic dependencies among the variables. An intervention on a subset $\boldsymbol{X} \subset \boldsymbol{V}$, denoted by $do(\boldsymbol{x})$, induces a sub-model $\mathcal{M}_{\boldsymbol{x}}$ in which $\boldsymbol{X}$ is set to constants $\boldsymbol{x}$, replacing the functions $\{f_X : X \in \boldsymbol{X}\}$ that would normally determine their values. The distribution of a set of variables $\boldsymbol{Y}$ in $\mathcal{M}_{\boldsymbol{x}}$ is denoted $P_{\boldsymbol{x}}(\boldsymbol{Y})$. Domains of $\boldsymbol{V}$ are discrete and finite.

An SCM induces a causal diagram $\mathcal{G}$ over $\boldsymbol{V}$, where $V \rightarrow W$ if $V$ appears as an argument of $f_W$, and $V \leftarrow\!\!\text{-}\!\!\text{-}\!\!\rightarrow W$ if $\boldsymbol{U}_V \cap \boldsymbol{U}_W \neq \varnothing$, ($V$ and $W$ share an unobserved confounder). In a causal diagram, two nodes are said to be in

---

[1]In practice, an assumption of strong faithfulness is typically required for consistently recovering (asymptotically, without error) the True PAG from finite samples (Robins et al., 2003; Zhang and Spirtes, 2012a).

[2]The criterion of d-separation follows (Pearl, 2009, Def. 1.2.3).

the same $c$-component $\boldsymbol{C} \subseteq \boldsymbol{V}$ if and only if they are connected by a bi-directed path, *i.e.*, a path composed entirely of edges "$\leftarrow\!\text{-}\!\text{-}\!\text{-}\!\rightarrow$". For any set $\boldsymbol{C} \subseteq \boldsymbol{V}$, $Q[\boldsymbol{C}] := P_{\boldsymbol{v}\backslash\boldsymbol{c}}(\boldsymbol{c})$ denotes the post-interventional distribution of $\boldsymbol{C}$ under an intervention on $\boldsymbol{V}\backslash\boldsymbol{C}$. By definition $Q[\boldsymbol{V}] = P(\boldsymbol{v})$ and by convention $Q[\varnothing] = 1$.

## 2 PROBLEM FORMULATION

In the setting of causal inference, we are interested in the following causal effect.

**Definition 1** (Causal effect)**.** *The causal effect from an intervention* $do(\boldsymbol{X} = \boldsymbol{x})$ *on an outcome* $\boldsymbol{Y}$ *is defined by* $P_{\boldsymbol{x}}(\boldsymbol{y})$.

The challenge is that we cannot immediately use this expression to estimate the causal effect as we only have access to the observational distribution $P$ but not the experimental distribution $P_{\boldsymbol{x}}$ that would define its value. In general, there might exist multiple SCMs $\mathcal{M}$ that entail the same data distribution $P(\boldsymbol{V})$ that result in different values of the causal effect $P_{\boldsymbol{x}}(\boldsymbol{y})$ (regardless of how many samples are collected). This motivates the problem of partial identification defined next.

**Definition 2** (Partial Identification)**.** *The causal effect* $P_{\boldsymbol{x}}(\boldsymbol{y})$ *is said to be partially identifiable from* $P(\boldsymbol{V})$ *if it determines a bound* $[a, b]$ *for* $P_{\boldsymbol{x}}(\boldsymbol{y})$ *that is strictly contained in* $[0, 1]$ *and valid over all SCMs* $\mathcal{M}$ *that induce* $P$.

We now introduce the so-called *natural bounds* (NB) due to Manski (1990); Robins (1989) that define a function of the observational data that consistently bounds $P_{\boldsymbol{x}}(\boldsymbol{y})$, irrespective of the causal structure of the system.

**Definition 3.** *The natural bounds (NBs) for a causal effect* $P_{\boldsymbol{x}}(\boldsymbol{y})$ *are given by,*

$$P(\boldsymbol{x}, \boldsymbol{y}) \leqslant P_{\boldsymbol{x}}(\boldsymbol{y}) \leqslant P(\boldsymbol{x}, \boldsymbol{y}) + 1 - P(\boldsymbol{x}). \quad (1)$$

In words, this result states that causal effect are naturally partially-identifiable. In particular, the NBs have been shown to be tight in several examples (in the sense that there exists two different models $\mathcal{M}^1, \mathcal{M}^2$ that entail $P(\boldsymbol{V})$ and evaluate to the lower and upper NBs, respectively). One example is the query $P_b(x)$ given $\mathcal{G}$ in Fig. 1a for which the NBs are tight.

For other queries that involve variables that are more "separated" in the underlying causal system, better bounds may be derived by exploiting the implications of "separation" on the entailed observational and interventional data distributions. For example, we would expect that if $(\boldsymbol{Z}\perp\!\!\!\perp\boldsymbol{Y})_P$ then $P_{\boldsymbol{x},\boldsymbol{z}}(\boldsymbol{y}) = P_{\boldsymbol{x}}(\boldsymbol{y})$ also and therefore the NBs could be improved. Statistical constraints of this type are an implication

of the structure of the underlying causal system onto the observed data with distribution $P(\boldsymbol{V})$. More generally, a $d$-separation between nodes in a causal diagram induces a corresponding conditional independence between variables in $\boldsymbol{V}$. The reverse implication, *i.e.* that each conditional independence in data implies a corresponding $d$-separation in the underlying causal diagram, is known as faithfulness. In particular, for three sets of variables $\boldsymbol{X}, \boldsymbol{Y}, \boldsymbol{Z}$ with a distribution $P(\boldsymbol{X}, \boldsymbol{Y}, \boldsymbol{Z})$ induced by a causal model with causal diagram $\mathcal{G}$, faithfulness asserts that,

$$(\boldsymbol{X}\perp\!\!\!\perp\boldsymbol{Y} \mid \boldsymbol{Z})_P \Rightarrow (\boldsymbol{X}\perp\!\!\!\perp\boldsymbol{Y} \mid \boldsymbol{Z})_{\mathcal{G}}.$$

This condition serves as a statistically testable constraint to narrow the class of compatible causal models (Pearl, 1988; Meek, 1995; Zhang, 2006). In this paper, we ask whether tighter bounds could be inferred under the assumption of faithfulness.

## 3 EQUIVALENCE CLASSES AND THEIR IMPLICATIONS

One common graphical abstraction to represent sets of causal diagrams with the same $d$-separation and non-ancestral relations are so called Maximal Ancestral Graphs (MAGs). "Ancestral" due to the fact that MAGs does not contain directed cycles (directed paths that start and end at the same node) or almost directed cycles (directed paths $X \rightarrow \cdots \rightarrow Y$ such that $X \leftarrow\!\text{-}\!\text{-}\!\text{-}\!\rightarrow Y$), and "maximal" due to the fact that every pair of nonadjacent nodes $\{X, Y\}$, there exists a set $\boldsymbol{Z} \subset \boldsymbol{V}$ that $d$-separates them.

Two causal diagrams or MAGs are said to be Markov equivalent (ME) if they entail the same set of $d$-separations[3]. A ME class of graphs can be summarized in a PAG that includes one additional edge tip "$\circ$" that denotes undecidability, *i.e.*, there are graphs in the equivalence class with both types of edge tips (Zhang, 2006, 2008a)[4]. Directed edges $X \rightarrow Y$ in a MAG or PAG are said to be visible, denoted $X \xrightarrow{v} Y$, if unobserved confounding can be ruled out. For example, the PAG in Fig. 1b encodes the ME class of the causal diagram in Fig. 1a. The output of the FCI algorithm is a PAG that can be recovered consistently under faithfulness (Spirtes et al., 2000; Zhang, 2006, 2008a).

**Notation.** It will be useful to use standard graph-theoretic family abbreviations to represent graphical relationships in causal diagrams or equivalence classes. A path between $X$ and $Y$ is potentially directed (causal) from $X$ to $Y$ if there is no arrowhead on the path pointing towards $X$. $Y$ is called a possible descendant of $X$, *i.e.*, $Y \in \texttt{PossDe}(X)$, and $X$ a possible ancestor of $Y$, *i.e.*, $X \in \texttt{PossAn}(Y)$, if there

---

[3]The notion corresponding to $d$-separation in MAGs is called $m$-separation.

[4]Selection bias, typically represented with undirected edges (Zhang, 2008b) or extra variables is not considered in this paper.

is a potentially directed path from $X$ to $Y$. By stipulation, $X \in \texttt{PossAn}(X)$. A set $\boldsymbol{X}$ is ancestral if no node outside $\boldsymbol{X}$ is a possible ancestor of any node in $\boldsymbol{X}$. $X$ is called a possible parent of $Y$, i.e., $X \in \texttt{PossPa}(Y)$, and $Y$ a possible child of $X$, i.e., $Y \in \texttt{PossCh}(X)$, if they are adjacent and the edge is not into $X$. Further, $X$ is called a possible spouse of $Y$, i.e., $X \in \texttt{PossSp}(Y)$, if they are adjacent and the edge is not visible. For a set of nodes $\boldsymbol{X}$, we have $\texttt{PossPa}(\boldsymbol{X}) = \bigcup_{X \in \boldsymbol{X}} \texttt{PossPa}(X)$. If the edge marks on a path between $X$ and $Y$ are all circles, we call the path a circle path. We refer to the closure of nodes connected with circle paths as a bucket. For example, in Fig. 1b $\{C, D\}$ is a bucket.

The notion of $pc$-component, defined below, generalizes that of $c$-components to equivalence classes and will be important for the proposed approach.

**Definition 4** ($pc$-component (Jaber et al., 2018a))**.** *In a PAG, or any induced sub-graph thereof, two nodes are in the same possible $c$-component ($pc$-component) if there is a path between them such that all non-endpoint nodes along the path are colliders, and none of the edges is visible.*

In words, the $pc$-component of a set $\boldsymbol{A}$ includes all the nodes which could, in some causal diagram, be in the $c$-component of some node in $\boldsymbol{A}$. Following this definition, e.g., $A, B$ and $X$ in Fig. 1b are in the same $pc$-component since $X$ is a collider on the path between them and none of the edges are visible. By contrast, $A$ and $C$ are not in the same $pc$-component since there is a visible edge on all paths that connect them. Using these notions, a causal effect of the form $Q[\boldsymbol{C}]$ can be decomposed into a product of smaller quantities, as shown in Prop. 1 using the Region construct.

**Definition 5** (Region Jaber et al. (2019a))**.** *Given a PAG $\mathcal{P}$ over $\boldsymbol{V}$, and $\boldsymbol{A} \subseteq \boldsymbol{C} \subseteq \boldsymbol{V}$. Let the region of $\boldsymbol{A}$ with respect to $\boldsymbol{C}$ be the union of the buckets that contain nodes in the $pc$-component of $\boldsymbol{A}$ in the sub-graph $\mathcal{P}_{\boldsymbol{C}}$.*

**Proposition 1** (Thm. 1 (Jaber et al., 2019a))**.** *Given a PAG $\mathcal{P}$ over $\boldsymbol{V}$ and $\boldsymbol{A} \subset \boldsymbol{C} \subseteq \boldsymbol{V}$, let the region of $\boldsymbol{A}$ with respect to $\boldsymbol{C}$ be denoted $\mathcal{R}_{\boldsymbol{A}}$. $Q[\boldsymbol{C}]$ can be decomposed as,*

$$ Q[\boldsymbol{C}] = Q[\mathcal{R}_{\boldsymbol{A}}] \cdot Q[\mathcal{R}_{\boldsymbol{C} \setminus \boldsymbol{A}}] \,/\, Q[\mathcal{R}_{\boldsymbol{A}} \cap \mathcal{R}_{\boldsymbol{C} \setminus \boldsymbol{A}}]. \quad (2) $$

Identification of quantities $Q[\cdot]$ given an equivalence class $\mathcal{P}$ uses a notion of (partial) topological order over the nodes in $\mathcal{P}$. A partial topological order is defined on buckets rather than individual nodes therefore extending the notion used in single causal diagrams and is valid for all causal diagrams in the Markov equivalence class (Jaber et al., 2018a, Lemma 1). For example, $A \prec B \prec X \prec \{C, D\}$, is a partial topological order over the buckets of $\mathcal{P}$ in Fig. 1b.

Conditional causal effects, of the form $P_{\boldsymbol{x}}(\boldsymbol{y} \mid \boldsymbol{z})$, can be similarly be decomposed using the notion of $Q[\cdot]$ by the definition of conditional probability,

$$ P_{\boldsymbol{x}}(\boldsymbol{y} \mid \boldsymbol{z}) = \sum_{\boldsymbol{c} \setminus \boldsymbol{y}} \left( Q[\boldsymbol{C} \cup \boldsymbol{Z}] \,/\, \sum_{\boldsymbol{c}} Q[\boldsymbol{C} \cup \boldsymbol{Z}] \right), \quad (3) $$

where $\boldsymbol{C} = \texttt{PossAn}(\boldsymbol{Y} \cup \boldsymbol{Z})_{\mathcal{P}_{\boldsymbol{V} \setminus \boldsymbol{x}}} \setminus \boldsymbol{X}$.

The decompositions of causal effects into $pc$-components and partial topological orders play a critical role in systematic identification algorithms and will be important in our work.

# 4 BOUNDING CAUSAL EFFECTS

This section aims to consider the separations between variables encoded in a PAG and the decomposition of causal effects it implies to provide a systematic algorithm to bound causal effects. After getting familiar with these decompositions our next task is to introduce a new notion of partial identification for this setting.

**Definition 6** (Partial Identification from a PAG)**.** *The causal effect $P_{\boldsymbol{x}}(\boldsymbol{y})$ is said to be partially identifiable from a PAG $\mathcal{P}$ and $P(\boldsymbol{v})$ if they determine a bound $[a, b]$ for $P_{\boldsymbol{x}}(\boldsymbol{y})$ that is strictly contained in $[0, 1]$ and is valid for any SCM compatible with $\mathcal{P}$.*

An SCM is said to be compatible or consistent with a PAG $\mathcal{P}$ if it induces a causal diagram that can be represented with $\mathcal{P}$. The following result shows that this notion of partial identification is driven by the constraints in the data distribution only, up to an assumption of faithfulness.

**Proposition 2.** *Let $\mathcal{P}$ be the PAG underlying $P(\boldsymbol{V})$. Under faithfulness, a causal effect is partially identifiable from $P(\boldsymbol{V})$ with bound $[a, b]$ if and only if it is partially identifiable from $\mathcal{P}$ and $P(\boldsymbol{V})$ with bound $[a, b]$.*

In words, Prop. 2 relates the solution space of two classes of models, namely the set of models compatible with a distribution $P(\boldsymbol{V})$ and the set of models compatible with $P(\boldsymbol{V})$ and the true PAG $\mathcal{P}$. It shows that the partial identification status of a query is preserved across settings under an assumption of faithfulness.

Our next results will be concerned with proposing a concrete procedure to derive bounds for the partial identification problem in Def. 6. The strategy involves bounding unidentifiable probabilities $Q[\boldsymbol{S}], \boldsymbol{S} \subset \boldsymbol{C}$ in terms of larger identifiable probabilities $Q[\boldsymbol{C}]$. These will then be introduced into existing identification algorithms from a PAG based on the decomposition in Prop. 1 to produce a systematic bounding algorithm.

**Proposition 3** (Lower bound)**.** *Given a PAG $\mathcal{P}$, consider sets $\boldsymbol{S} \subset \boldsymbol{C} \subseteq \boldsymbol{V}$ and define $\boldsymbol{W} = \texttt{PossAn}(\boldsymbol{S})_{\mathcal{P}_{\boldsymbol{C}}}$, $\boldsymbol{R} = \boldsymbol{W} \setminus \boldsymbol{S}$, and $\boldsymbol{T} = \texttt{PossSp}(\boldsymbol{S})_{\mathcal{P}_{\boldsymbol{C}}} \setminus \boldsymbol{S}$. Let $\boldsymbol{A}, \boldsymbol{B}$ partition $\boldsymbol{R}$*

such that $\boldsymbol{B} = PossDe(\boldsymbol{T})_{\mathcal{P}_C} \cap \boldsymbol{R}, \boldsymbol{A} = \boldsymbol{R} \backslash \boldsymbol{B}$. $Q[\boldsymbol{S}]$ is lower bounded as follows:

$$Q[\boldsymbol{S}] \geqslant \max_{\boldsymbol{z}} \frac{Q[\boldsymbol{W}]}{\sum_{\boldsymbol{s},\boldsymbol{b}} Q[\boldsymbol{W}]}, \tag{4}$$

where $\boldsymbol{Z} = PossPa(\boldsymbol{W})_{\mathcal{P}} \backslash PossPa(\boldsymbol{S})_{\mathcal{P}}$.

**Proposition 4** (Upper bound). *Given a PAG $\mathcal{P}$, consider sets $\boldsymbol{S} \subset \boldsymbol{C} \subseteq \boldsymbol{V}$ and let a partial topological ordering of $\boldsymbol{S}$ be $\boldsymbol{S}_1 \prec \cdots \prec \boldsymbol{S}_k$. Define $\boldsymbol{W} = PossAn(\boldsymbol{S})_{\mathcal{P}_C}$, $\boldsymbol{R} = \boldsymbol{W} \backslash \boldsymbol{S}$, and $\boldsymbol{T} = PossSp(\boldsymbol{S})_{\mathcal{P}_C} \backslash \boldsymbol{S}$. Let $\boldsymbol{A}, \boldsymbol{B}$ partition $\boldsymbol{R}$ such that $\boldsymbol{B} = PossDe(\boldsymbol{T})_{\mathcal{P}_C} \cap \boldsymbol{R}, \boldsymbol{A} = \boldsymbol{R} \backslash \boldsymbol{B}$. $Q[\boldsymbol{S}]$ is upper bounded as follows:*

$$Q[\boldsymbol{S}] \leqslant \min_{\boldsymbol{z}} \left\{ \frac{Q[\boldsymbol{W}]}{\sum_{\boldsymbol{s},\boldsymbol{b}} Q[\boldsymbol{W}]} - \sum_{\boldsymbol{s}_k} \frac{Q[\boldsymbol{W}]}{\sum_{\boldsymbol{s},\boldsymbol{b}} Q[\boldsymbol{W}]} \right\} + Q[\boldsymbol{S} \backslash \boldsymbol{S}_k], \tag{5}$$

where $\boldsymbol{Z} = PossPa(\boldsymbol{W})_{\mathcal{P}} \backslash PossPa(\boldsymbol{S})_{\mathcal{P}}$.

These results use graph theoretic notation to distinguish between qualitatively different relationships among variables in a PAG. The following example illustrates these results more concretely.

**Example 1** (Contrast with natural bounds). Consider the evaluation of a query $P_b(x)$ given the distribution $P(x,b)$ that does not advertise any statistical independencies between $X$ and $B$. The corresponding PAG is given by $\{B \circ\!\!-\!\!\circ X\}$. With the notation of the propositions above, $P_b(x) = Q[\boldsymbol{S}], \boldsymbol{S} = \{X\}, \boldsymbol{W} = \{B, X\}, \boldsymbol{B} = \{B\}, \boldsymbol{Z} = \varnothing, \boldsymbol{S}_k = \boldsymbol{S}$, and therefore,

$$P(b,x) \leqslant P_b(x) \leqslant P(b,x) - P(b) + 1, \tag{6}$$

using the facts that $Q[\boldsymbol{W}] = P(b,x), \sum_{\boldsymbol{s}} Q[\boldsymbol{W}] = P(b)$. These expressions recover the NBs. With additional independencies, tighter bounds could be given by the proposed techniques. Continuing with this example, assume that in addition to $X, B$ we observe samples from variables $A, D, C$ whose conditional independencies are summarized by the PAG in Fig. 1a. Consider now the query $P_{b,d,c}(x,a)$. By inspecting the PAG we find that $\boldsymbol{S} = \{X, A\}, \boldsymbol{W} = \{A, B, X\}, \boldsymbol{B} = \{A, B\}, \boldsymbol{S}_k = \{X\}, \boldsymbol{Z} = \varnothing$, and,

$$P(b,x,a) \leqslant P_{b,d,c}(x,a) \leqslant P(b,x,a) - P(a,b) + P(a). \tag{7}$$

In contrast, the NBs return $P(b,x,a,c,d) \leqslant P_{b,d,c}(x,a) \leqslant P(b,x,a,c,d) - P(b,c,d) + 1$. We can verify that the proposed lower bound is tighter as

$$(\text{NB} =) \quad P(b,x,a,c,d) \leqslant P(b,x,a). \tag{8}$$

And, the upper bound is tighter as,

$$\begin{aligned}
(\text{NB} =) \quad & P(b,x,a,c,d) - P(b,c,d) + 1 \\
& \overset{(1)}{\geqslant} \sum_{c,d} \left\{ P(b,x,a,c,d) - P(b,c,d) \right\} + 1 \\
& = P(b,x,a) - P(b) + 1 \\
& = P(b,x,a) + \sum_a \left\{ P(a) - P(a,b) \right\} \\
& \overset{(2)}{\geqslant} P(b,x,a) + P(a) - P(a,b). \tag{9}
\end{aligned}$$

The first and last inequalities (1,2) hold by noting that $P(b,x,a,c,d) - P(b,c,d) \leqslant 0$ and $P(a) - P(a,b) \geqslant 0$, respectively. ∎

We can see that in some cases these bounds coincide with the NBs, as in Eq. (6), while in others they improve upon the NBs, as in Eqs. (8) and (9). The following result makes this claim more concrete.

**Proposition 5.** *Consider a query $P_{\boldsymbol{x}}(\boldsymbol{y})$ and let $\mathcal{P}$ be the PAG over $\{\boldsymbol{X}, \boldsymbol{Y}\}$ compatible with $P$. Then, under an assumption of faithfulness, the bounds given in Props. 3 and 4 are at least as tight as the natural bounds.*

It is worth emphasizing that this result does not come for free. The assumption of faithfulness is critical: without it, no $d$-separation could be guaranteed, the compatible PAG would have to be fully connected (and non-informative), and consequently the proposed bounds would revert to the NBs in all cases.

We are now ready to describe a systematic algorithm for bounding arbitrary causal effects that exploits the new bounds. The procedure is given in Alg. 1. It extends the identification algorithm IDP (Jaber et al., 2019a) with a call to the propositions above (line 12) whenever a component $Q[\cdot]$ is not uniquely identifiable.

**Proposition 6.** *Partial IDP (Alg. 1) terminates and is sound.*

The proof follows from the soundness of IDP (Jaber et al., 2019a) and Props. 3 and 4. A similar algorithm for partially identifying conditional causal effects could be derived by adapting CIDP (conditional IDP) due to Jaber et al. (2022) with a call to the propositions above whenever a component is not uniquely identifiable. To get more familiar with the proposed procedure, we exemplify the various steps of Partial IDP in several additional scenarios.

**Example 2** (Steps of Partial IDP). Consider the query $P_{x,w,z}(y)$ given $\mathcal{P}$ in Fig. 2. Notice that the effect is not immediately identifiable as the path $Z \rightarrow Y$ start with an invisible edge (that doesn't rule out unobserved confounding between $Z$ and $Y$ (Jaber et al., 2019a, Thm. 3). The first step in lines 1 and 2 of Alg. 1 involves identifying

**Algorithm 1** Partial IDP

**Input:** A PAG $\mathcal{P}$ and disjoint sets $\boldsymbol{X}, \boldsymbol{Y} \subset \boldsymbol{V}$
**Output:** Lower or upper bound expressions ($type$) for
$\quad P_{\boldsymbol{x}}(\boldsymbol{y})$
1: Let $\boldsymbol{D} := \texttt{PossAn}(\boldsymbol{Y})_{\mathcal{P}_{\boldsymbol{V} \setminus \boldsymbol{X}}}$
2: **return** $\sum_{\boldsymbol{d} \setminus \boldsymbol{y}} \texttt{PID}(\boldsymbol{D}, \boldsymbol{V}, P, type)$

3: **function** $\texttt{PID}(\boldsymbol{C}, \boldsymbol{T}, Q = Q[\boldsymbol{T}], type)$
4: $\quad$ **if** $\boldsymbol{C} = \varnothing$ then return 1.
5: $\quad$ **if** $\boldsymbol{C} = \boldsymbol{T}$ then return $Q$.
$\quad$ /* In $\mathcal{P}_{\boldsymbol{T}}$, let $\boldsymbol{B}$ denote a bucket, and let $\boldsymbol{C_B}$ denote
$\quad$ the $pc$-component of $\boldsymbol{B}$ */
6: $\quad$ **if** $\exists \boldsymbol{B} \subset \boldsymbol{T} \setminus \boldsymbol{C}$ such that $\boldsymbol{C_B} \cap \texttt{PossCh}(\boldsymbol{B})_{\mathcal{P}_{\boldsymbol{T}}} \subseteq \boldsymbol{B}$
$\quad$ **then**
7: $\quad\quad$ Compute $Q[\boldsymbol{T} \setminus \boldsymbol{B}]$ from $Q[\boldsymbol{T}]$ via (Jaber et al.,
$\quad\quad$ 2018a, Prop. 2).
8: $\quad\quad$ **return** $\texttt{PID}(\boldsymbol{C}, \boldsymbol{T} \setminus \boldsymbol{B}, Q[\boldsymbol{T} \setminus \boldsymbol{B}], type)$
9: $\quad$ **else if** $\exists \boldsymbol{B} \subset \boldsymbol{C}$ such that $\mathcal{R}_{\boldsymbol{B}} \neq \boldsymbol{C}$ **then**
10: $\quad\quad$ **if** Lower bound desired **then**
11: $\quad\quad\quad$ **return** $\frac{\texttt{PID}(\mathcal{R}_{\boldsymbol{B}}, \boldsymbol{T}, Q, lower) \cdot \texttt{PID}(\mathcal{R}_{\boldsymbol{C} \setminus \mathcal{R}_{\boldsymbol{B}}}, \boldsymbol{T}, Q, lower)}{\texttt{PID}(\mathcal{R}_{\boldsymbol{B}} \cap \mathcal{R}_{\boldsymbol{C} \setminus \mathcal{R}_{\boldsymbol{B}}}, \boldsymbol{T}, Q, upper)}$
12: $\quad\quad$ **else**
13: $\quad\quad\quad$ **return** $\frac{\texttt{PID}(\mathcal{R}_{\boldsymbol{B}}, \boldsymbol{T}, Q, upper) \cdot \texttt{PID}(\mathcal{R}_{\boldsymbol{C} \setminus \mathcal{R}_{\boldsymbol{B}}}, \boldsymbol{T}, Q, upper)}{\texttt{PID}(\mathcal{R}_{\boldsymbol{B}} \cap \mathcal{R}_{\boldsymbol{C} \setminus \mathcal{R}_{\boldsymbol{B}}}, \boldsymbol{T}, Q, lower)}$
14: $\quad\quad$ **end if**
15: $\quad$ **else**
16: $\quad\quad$ **return** Lower or upper bounds (according to $type$)
$\quad\quad$ for $Q[\boldsymbol{C}]$ from $Q[\boldsymbol{T}]$ via Props. 3 and 4.
17: $\quad$ **end if**

$Q[\boldsymbol{D}], \boldsymbol{D} = \texttt{PossAn}(Y)_{\mathcal{P}_{\{S,Y\}}} = \{Y\}$ by running the
IDP procedure in line 3. The first if condition, on line
6, is triggered as $\boldsymbol{B} = \{S\} \subset \boldsymbol{T} \setminus \boldsymbol{C} = \{W, X, Z, S\}$
satisfies that $\boldsymbol{C}_S \cap \texttt{PossCh}(S)_{\mathcal{P}_{\boldsymbol{T}}} = \varnothing$ that is trivially
included in $S$. Following (Jaber et al., 2018a, Prop. 2),
in line 7 we can therefore evaluate $Q[W, X, Z, Y]$ from
$Q[\boldsymbol{T}] = P(w, x, z, y, s)$ that returns $Q[W, X, Z, Y] =
P(w, x, y, z)$. Next we consider applying $\texttt{PID}$ with the
set $\boldsymbol{T} = \{W, X, Z, Y\}$, finding that $\boldsymbol{B} = \{X, W\}$ trig-
gers the if condition, as the intersection of $\boldsymbol{C}_{\{X,W\}} =
\{W, X, Z\}$ and $\texttt{PossCh}(\{X, W\})_{\mathcal{P}_{\{W,X,Z,Y\}}} = \{X, Y\}$
equals $\{W, X\}$ which is included in $\boldsymbol{B}$. This licenses
the evaluation of $Q[Z, Y]$ from $Q[W, X, Z, Y]$ to obtain
$Q[Z, Y] = P(y \mid z, x) P(z)$, further simplifying the set
$\boldsymbol{T} = \{Z, Y\}$. The next call to $\texttt{PID}$ reveals that none of the
if conditions in lines 6 and 9 are triggered and we have to
resort to the computation of bounds on $Q[Y]$ from $Q[Y, Z]$
in line 16. A call to Props. 3 and 4 then returns:

$$Q[Y] \geqslant Q[Y, Z] = P(y \mid z, x) P(z), \qquad (10)$$

which implies $P_{x,w,z}(y) \geqslant P(y \mid z, x) P(z)$. For the upper
bound,

$$Q[Y] \leqslant Q[Y, Z] - \sum_y Q[Y, Z] + 1, \qquad (11)$$

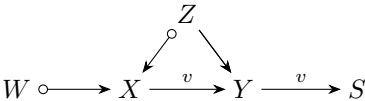

Figure 2: PAG for Example 2.

which implies $P_{x,w,z}(y) \leqslant P(y \mid z, x) P(z) - P(z) + 1$.

We could show, moreover, that these bounds are tighter than
the NBs as, for the lower bound,

$$
\begin{aligned}
(\text{NB} =) \quad & P(y, z, x, w) \\
& \leqslant P(y, z, x) \\
& = P(y \mid z, x) P(x \mid z) P(z) \\
& \stackrel{(1)}{\leqslant} P(y \mid z, x) P(z), \qquad (12)
\end{aligned}
$$

which is the proposed lower bound. (1) holds because $P(x \mid
z) \leqslant 1$. For the upper bound,

$$
\begin{aligned}
(\text{NB} =) \quad & P(y, z, x, w) - P(z, x, w) + 1 \\
& \stackrel{(2)}{\geqslant} 1 + \sum_w \big\{ P(y, z, x, w) - P(z, x, w) \big\} \\
& = 1 + P(x \mid z) \big\{ P(y \mid z, x) P(z) - P(z) \big\} \\
& \stackrel{(3)}{\geqslant} P(y \mid z, x) P(z) - P(z) + 1, \qquad (13)
\end{aligned}
$$

which is the proposed upper bound. (2) holds because
$P(y, z, x, w) - P(z, x, w) \leqslant 0$ and (3) holds because
$P(x \mid z) \leqslant 1$ and the term in brackets $\{\cdot\} \leqslant 0$. $\blacksquare$

**Example 3** (Applications in biology). We illustrate next
the inference that could be made for decision making in
medicine and healthcare with a (publicly available) dataset
from the literature.

We revisit the protein signalling study of (Sachs et al., 2005).
Signalling pathways regulate the activity of a cell. The abil-
ity to precisely predict the effect of perturbations in sig-
nalling pathways, *e.g.*, by knocking out a gene that inacti-
vates a protein in the network, on a phenotype of interest,
such as cell growth, can have important applications for the
treatment of disease. We consider computing bounds on the
effect of PKC inactivation on the RAF/MEK/ERK pathway
given the PAG in Fig. 3b recovered from phosphorylation
data (*i.e.* markers of pathway activation)[5].

In this example, the query of interest is given by
$P(\text{RAF}, \text{MEK}, \text{ERK} \mid do(\text{PKC}))$. In line 2 of Alg. 1, we
may rewrite this quantity as $\sum_{\text{PKA}} Q[\text{RAF}, \text{MEK}, \text{ERK}, \text{PKA}]$
where we have used the ancestral set of the outcome vari-
ables. A call to $\texttt{PID}$ then triggers the if condition in line
9 in which the bucket $\boldsymbol{B} = \{\text{MEK}\}$ for which the region

---

[5]We use a discretized version of the data following (Hartemink
et al., 2000) with levels: high (2), average (1), low (0) activation,
downloaded from the `bnlearn` data repository (Scutari, 2009).

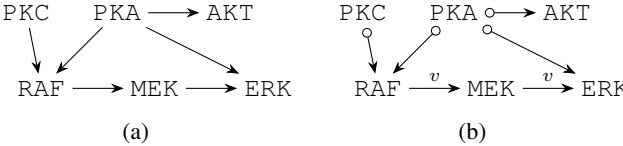

Figure 3: (a) Protein signalling network (Sachs et al., 2005, Fig. 2), (b) corresponding PAG for Example 3.

$\mathcal{R}_{\boldsymbol{B}} = \{\texttt{MEK}\} \neq \{\texttt{RAF}, \texttt{MEK}, \texttt{ERK}, \texttt{PKA}\}$. We may therefore decompose $Q[\texttt{RAF}, \texttt{MEK}, \texttt{ERK}, \texttt{PKA}]$ into two terms $Q[\texttt{RAF}, \texttt{ERK}, \texttt{PKA}]$ and $Q[\texttt{MEK}]$ that may be considered separately. Among these expressions: $Q[\texttt{MEK}] = P(\texttt{MEK} \mid \texttt{RAF})$ is identifiable but $Q[\texttt{RAF}, \texttt{ERK}, \texttt{PKA}]$ isn't. Calling IPD on the second term we find, however, that the first if condition in line 6 is triggered: $\boldsymbol{T} = \boldsymbol{V}$ reduces to $\boldsymbol{T} = \{\texttt{RAF}, \texttt{ERK}, \texttt{PKA}, \texttt{PKC}\}$ and

$$Q[\texttt{RAF}, \texttt{ERK}, \texttt{PKA}, \texttt{PKC}] = \tag{14}$$
$$P(\texttt{RAF}, \texttt{ERK}, \texttt{PKA}, \texttt{PKC}, \texttt{MEK})/P(\texttt{MEK} \mid \texttt{RAF})$$

Finally, we now proceed to bound $Q[\texttt{RAF}, \texttt{ERK}, \texttt{PKA}]$ from $Q[\texttt{RAF}, \texttt{ERK}, \texttt{PKA}, \texttt{PKC}]$ with Props. 3 and 4. Following the notation of Props. 3 and 4, in $\mathcal{P}_{\{\texttt{RAF}, \texttt{ERK}, \texttt{PKA}, \texttt{PKC}\}}$, $\boldsymbol{W} = \texttt{PossAn}(\boldsymbol{S}) = \{\texttt{RAF}, \texttt{ERK}, \texttt{PKA}, \texttt{PKC}\}, \boldsymbol{R} = \boldsymbol{B} = \{\texttt{PKC}\}$, and $\boldsymbol{Z} = \varnothing$. It then follows that,

$$Q[\texttt{RAF}, \texttt{ERK}, \texttt{PKA}] \geqslant Q[\texttt{RAF}, \texttt{ERK}, \texttt{PKA}, \texttt{PKC}], \tag{15}$$

and that,

$$Q[\texttt{RAF}, \texttt{ERK}, \texttt{PKA}] \leqslant Q[\texttt{RAF}, \texttt{ERK}, \texttt{PKA}, \texttt{PKC}] \tag{16}$$
$$- \sum_{\texttt{ERK}} Q[\texttt{RAF}, \texttt{ERK}, \texttt{PKA}, \texttt{PKC}] + Q[\texttt{RAF}, \texttt{PKA}].$$

$Q[\texttt{RAF}, \texttt{PKA}]$ on the r.h.s. could be further upper-bounded from $Q[\texttt{RAF}, \texttt{PKA}, \texttt{PKC}]$ with a similar strategy; in particular giving $Q[\texttt{RAF}, \texttt{PKA}] \leqslant Q[\texttt{RAF}, \texttt{PKA}, \texttt{PKC}] + 1 - Q[\texttt{PKC}] = P(\texttt{RAF}, \texttt{PKA}, \texttt{PKC}) + 1 - P(\texttt{PKC})$.

Combining these expressions, we could use the observed data to estimate the conditionals and infer the probabilities of high and low pathway activation after knocking out (intervening on) PKC:

$$P(\texttt{RAF} = 0, \texttt{MEK} = 0, \texttt{ERK} = 0 \mid do(\texttt{PKC} = 0)) \tag{17}$$
$$\in [0.0214, 0.0864],$$
$$P(\texttt{RAF} = 2, \texttt{MEK} = 2, \texttt{ERK} = 2 \mid do(\texttt{PKC} = 0)) \tag{18}$$
$$\in [0.1120, 0.3115],$$

respectively. In turn, by relying on the current consensus biological network (Sachs et al., 2005, Fig. 2), given in Fig. 3a, the causal effects would be point estimated to be 0.0441 and 0.1861 respectively. Without committing to a particular causal diagram, the inferred bounds would be the more cautious approximation of causal effects. ∎

We provide additional worked examples to illustrate the proposed bounding technique in Appendix C.

**Remark** (Statistical uncertainty). The bounds computed in Example 3 do not account for statistical uncertainty (both in the discovery of the PAG and in the approximation of bounds). In practice, an assumption of strong faithfulness is typically required for consistently recovering the True PAG from finite samples (Robins et al., 2003; Zhang and Spirtes, 2012a). A more careful analysis would be required to make actionable causal claims.

For some queries, we could show that the bounds returned by Partial IDP in Alg. 1 are tight: one example is the first query in Example 1 (Eq. (6)). In general, however, analytical bounds returned for arbitrary queries and equivalence classes are not known to be tight. Determining whether this is the case is an important research direction.

In the next section, to investigate the derivation of tighter bounds for arbitrary queries, we switch gears and consider a different approach to bounding causal effects in equivalence classes, namely enumeration strategies.

## 5 THE DIFFICULTY OF ENUMERATING CAUSAL DIAGRAMS FROM A PAG

The techniques explored so far overlook the potential for enumerating (relevant) ME causal diagrams and subsequently applying existing (partial) identification techniques given each diagram separately (that we refer to as "enumeration strategies"). Enumerating all ME causal diagrams is exponentially costly, and intractable in general even with an assumption of no unobserved confounding, *i.e.* in the space of directed acyclic graphs as shown by Wienöbst et al. (2023). However, a number of observations can be made to avoid enumerating all ME causal diagrams which reduces the search space to a (potentially tractable) *subset* of "relevant" ME causal diagrams without loss of generality.

This section explores the definition of sets of ME causal diagrams $\mathcal{K} \subset \mathcal{P}$ with the distinctiveness of being equally expressive in the sense that,

$$\left\{ P_{\boldsymbol{x}}(\boldsymbol{y}; \mathcal{M}) : \mathcal{M} \in \mathbb{M}(\mathcal{P}) \right\} =$$
$$\left\{ P_{\boldsymbol{x}}(\boldsymbol{y}; \mathcal{M}) : \mathcal{M} \in \mathbb{M}(\mathcal{K}) \right\}. \tag{19}$$

Let $\mathbb{M}(\mathcal{P})$ denote the set of SCMs compatible with $\mathcal{P}$, that is the set of SCMs that induce causal diagrams contained in the PAG abstraction $\mathcal{P}$. Under Eq. (19), minimum and maximum values of causal effects remain unchanged, and one may exploit $\mathcal{K}$ instead of $\mathcal{P}$ for inference in practice. The hope is that if the set of causal diagrams $\mathcal{K}$ is small enough then we might be able to apply existing partial identification algorithms on every causal diagram $\mathcal{G} \in \mathcal{K}$ efficiently.

We start by introducing the notion of a Loyal Equivalent

$$X \circ\!\!\rightarrow Y \leftarrow\!\!\circ Z \qquad X \overset{\curvearrowright}{\phantom{x}} Y \leftarrow Z \qquad X \rightarrow Y \leftarrow Z \qquad X \overset{\curvearrowright}{\rightarrow} Y \overset{\curvearrowleft}{\leftarrow} Z$$

(a) PAG     (b) MAG     (c) LEG     (d) MBD

Figure 4: Diagrams used in Sec. 5.

Graph (LEG) (Def. 7), from (Zhang and Spirtes, 2012b, Prop. 2), that are sets of ME MAGs that retain "expressiveness" in the sense of Eq. (19). This result is given in Prop. 7.

**Definition 7.** *Given a MAG $\mathcal{G}$, there exists a ME MAG $\mathcal{H}$, called a Loyal Equivalent Graph, such that all bi-directed edges in $\mathcal{H}$ are invariant, and every directed edge in $\mathcal{G}$ is also in $\mathcal{H}$.*

**Proposition 7** (Expressiveness of LEGs)**.** *Given a PAG $\mathcal{P}$, let $\mathcal{L}$ be the set of ME LEGs. Then, $\mathbb{M}(\mathcal{P}) = \mathbb{M}(\mathcal{L})$.*

For example, the LEG in Fig. 4c is derived from the MAG in Fig. 4b by replacing the bi-directed edge with a directed one. Prop. 7 shows that the set of ME LEGs is as expressive as the set of ME MAGs that are encoded by $\mathcal{P}$[6]. The significance of this proposition lies in the fact that ME LEGs are a subset of ME MAGs and that, contrary to ME MAGs, ME LEGs are in principle listable by exhaustively applying a simple criterion for the reversal of directed edges while remaining Markov equivalent, *i.e.* (Zhang and Spirtes, 2012b, Lemma 2). We review this criterion in more detail in Appendix A and give an algorithm for enumerating ME LEGs that exploits it in Appendix D.

A second redundancy result is given in Prop. 8 by introducing so called *maximally bi-directed* (MBD) diagrams.

**Definition 8.** *A causal diagram $\mathcal{G}$ is said to be maximally bi-directed if no further bi-directed edges can be added without breaking a $d$-separation.*

**Proposition 8** (Expressiveness of MBD diagrams)**.** *Given a PAG $\mathcal{P}$, let $\mathcal{D}$ be the set of ME MBD diagrams. Then, $\mathbb{M}(\mathcal{P}) = \mathbb{M}(\mathcal{D})$.*

MDB causal diagrams can be constructed from an LEG by adding bi-directed edges on top of invisible directed edges wherever possible. For example, the causal diagram $\mathcal{G}$ in Fig. 4d, compatible with the LEG $L$ in Fig. 4c, is said to be maximally bi-directed as no further bi-directed edges can be added while remaining Markov equivalent. $\mathcal{G}$ has the distinctiveness of inducing a family of SCMs which includes all SCMs that are compatible with any causal diagram compatible with the corresponding LEG $L$ in Fig. 4c, i.e. $\mathbb{M}(\mathcal{G}) = \mathbb{M}(L)$. More specifically, in this example, $\mathcal{G}$ induces a class of SCMs given by: $x := f_X(\boldsymbol{u}_{X,Y}), z :=$

---

[6]Recall, as noted in Sec. 3, that MAGs (and LEGs) encode the sets of causal diagrams that would be represented by the same MAG (or LEG).

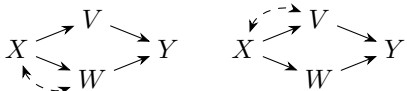

Figure 5: Diagrams used in Prop. 9.

$f_Z(\boldsymbol{u}_{X,Z}), y := f_Y(x, z, \boldsymbol{u}_{X,Z}, \boldsymbol{u}_{X,Y})$, with deterministic functions and exogeneous distributions arbitrarily defined. We can see that this parameterization is flexible enough to represent any SCM induced by causal diagrams compatible with $L$. As a consequence, in general, Prop. 8 implies that the set of ME MBD causal diagrams is as expressive as the set of *all* ME causal diagrams. Appendix D gives an algorithm for generating all ME MBD diagrams $\mathcal{D}$ from the set of all ME LEGs $\mathcal{L}$.

Additionally, note that in general multiple MBD causal diagrams can be derived from with a single LEG. For example, Fig. 5 gives two MBD causal diagrams induced by the same LEG (both bi-directed edges could not appear simultaneously as that would violate the independence $(V \perp\!\!\!\perp W \mid X)_P$). Therefore, unfortunately, a reduction of the set of ME MBD diagrams $\mathcal{D}$ while preserving the space of SCMs $\mathbb{M}(\mathcal{D})$ is, in general, not possible. Multiple MBD diagrams for each LEG may have to be considered to properly characterize causal effects given a PAG $\mathcal{P}$.

**Proposition 9** (Non-redundancy)**.** *Given a PAG $\mathcal{P}$, let $\mathcal{G}$ and $\mathcal{H}$ be two MBD diagrams constructed from a ME LEG. In general, $\mathbb{M}(\mathcal{G}) \nsubseteq \mathbb{M}(\mathcal{H})$ and $\mathbb{M}(\mathcal{H}) \nsubseteq \mathbb{M}(\mathcal{G})$.*

In other words, bounds for a given causal effect computed given two MBD diagrams will in general be different and one has to consider both diagrams to correctly characterize bounds on causal effects given an equivalence class. The proof proceeds by exploiting the MBD causal diagrams in Fig. 5 as a counter-example. As a consequence of this result, we conjecture that enumeration techniques, in the worst-case, would have to consider all MBD diagrams separately.

To better understand the computation cost of enumerating LEGs and MBD causal diagrams, we propose the first enumeration algorithms for this purpose in Alg. 3 and Alg. 4 in Appendix D (proceeding similarly to how one might enumerate Markov equivalent DAGs as done by Wienöbst et al. (2023)). These (potentially sub-optimal) procedures suggest that doing enumerating "relevant" causal diagrams requires a polynomial cost in the number of edges of LEGs, in addition to the computational cost of the bounding algorithms themselves[7]. Overall, this analysis suggests that existing

---

[7]For example, in Zhang et al. (2021), inference of bounds re-

bounding techniques, even with the consideration of redundancies presented in this section, are not practical beyond a handful of variables (Zeitler and Silva, 2022). Still, given that the proposed bounds (Sec. 4) have not been shown to be tight in general, one might still be interested in enumerating the MBD causal diagrams to get more informative bounds, despite computational costs.

# 6  CONCLUSIONS

Causal effect estimation is a common inference problem across different applications, many of which do not have a known causal diagram describing the system of variables. In this paper, we study the problem of partial identification with knowledge of a Markov equivalence class, represented by a Partial Ancestral Graph (PAG), that can be inferred from data. We demonstrate that analytical bounds can be derived by exploiting the invariances present in the PAG and the corresponding decomposition of causal effects. These are the first bounds on causal effects in the literature that exploit the knowledge encoded in a PAG. We further consider enumeration techniques, that list relevant causal diagrams and apply partial identification techniques on each one separately. We show that despite several redundancies within the space of ME causal diagrams, the computational cost likely remains large in practice.

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

# Appendix

This Appendix includes

- Additional background, related work, and a discussion on the limitations of the proposed approach in Appendix A.
- Proofs in Appendix B.
- Additional examples in Appendix C.
- Further discussion on enumeration strategies, including algorithms for enumerating LEGs and MDB causal diagrams, in Appendix D.

## A  BACKGROUND, RELATED WORK, AND LIMITATIONS

### A.1  BACKGROUND

In this section, we review several graphical notions that are used in the main body of this document or will be used for the derivation of proofs.

Firstly, we review manipulations in causal diagrams $\mathcal{G}$. Let $\mathcal{G}$ denote a causal diagram over $\boldsymbol{V}$ and $\boldsymbol{X} \subseteq \boldsymbol{V}$. $\mathcal{G}_{\boldsymbol{X}}$ denotes the induced subgraph of $\mathcal{G}$ over $\boldsymbol{X}$. The $\boldsymbol{X}$-lower-manipulation of $\mathcal{G}$ deletes all those edges that are out of variables in $\boldsymbol{X}$ and otherwise keeps $\mathcal{G}$ as it is. The resulting graph is denoted as $\mathcal{G}_{\underline{\boldsymbol{X}}}$. The $\boldsymbol{X}$-upper-manipulation of $\mathcal{G}$ deletes all those edges in $\mathcal{G}$ that are into variables in $\boldsymbol{X}$, and otherwise keeps $\mathcal{G}$ as it is. The resulting graph is denoted as $\mathcal{G}_{\overline{\boldsymbol{X}}}$. Further, we will use standard graph-theoretic family abbreviations to represent graphical relationships in graphs $\mathcal{G}$, such as parents $pa$, descendants $de$, ancestors $an$, and spouses $sp$. For example, let $X \in sp(Y)_{\mathcal{G}}$ if $X \leftarrow\!\!-\!\!-\!\!-\!\!\rightarrow Y$ is present in $\mathcal{G}$. Capitalized versions $Pa, De, An, Sp$ include the argument as well, e.g. $Pa(\boldsymbol{X})_{\mathcal{G}} = pa(\boldsymbol{X})_{\mathcal{G}} \cup \boldsymbol{X}$. If $X \in an(Y)_{\mathcal{G}} \cap sp(Y)_{\mathcal{G}}$, we say that there is an almost directed cycle between $X$ and $Y$.

The following rules to manipulate experimental distributions produced by an intervention are known as the do-calculus and will be used for the proof of several theoretical statements (Pearl, 2009).

**Theorem 1** (Inference Rules $do$-calculus). *Let $\mathcal{G}$ be a causal diagram compatible with an SCM $\mathcal{M}$, with endogenous variables $\boldsymbol{V}$. For any disjoint subsets $\boldsymbol{X}, \boldsymbol{Y}, \boldsymbol{Z} \subseteq \boldsymbol{V}$, two disjoint subsets $\boldsymbol{Z}, \boldsymbol{W} \subseteq \boldsymbol{V} \backslash (\boldsymbol{Y} \cup \boldsymbol{X})$, the following rules are valid for any intervention strategies $do(\boldsymbol{X} = \boldsymbol{x}), do(\boldsymbol{Z} = \boldsymbol{z})$:*

- *Rule 1 (Insertion/Deletion of observations):*

$$P_{\boldsymbol{x}}(\boldsymbol{y} \mid \boldsymbol{w}, \boldsymbol{z}) = P_{\boldsymbol{x}}(\boldsymbol{y} \mid \boldsymbol{w}) \quad \textit{if} \quad (\boldsymbol{Z} \perp\!\!\!\perp \boldsymbol{Y} \mid \boldsymbol{W}, \boldsymbol{X})_{\mathcal{G}_{\overline{\boldsymbol{X}}}}.$$

- *Rule 2 (Change of regimes):*

$$P_{\boldsymbol{x}, \boldsymbol{z}}(\boldsymbol{y} \mid \boldsymbol{w}) = P_{\boldsymbol{x}}(\boldsymbol{y} \mid \boldsymbol{z}, \boldsymbol{w}) \quad \textit{if} \quad (\boldsymbol{Y} \perp\!\!\!\perp \boldsymbol{Z} \mid \boldsymbol{W}, \boldsymbol{X})_{\mathcal{G}_{\overline{\boldsymbol{X}}, \underline{\boldsymbol{Z}}}}.$$

- *Rule 3 (Insertion/Deletion of interventions):*

$$P_{\boldsymbol{x}, \boldsymbol{z}}(\boldsymbol{y} \mid \boldsymbol{w}) = P_{\boldsymbol{x}}(\boldsymbol{y} \mid \boldsymbol{w}) \quad \textit{if} \quad (\boldsymbol{Y} \perp\!\!\!\perp \boldsymbol{Z} \mid \boldsymbol{W}, \boldsymbol{X})_{\mathcal{G}_{\overline{\boldsymbol{X}, \boldsymbol{Z}(\boldsymbol{W})}}}.$$

*where $\boldsymbol{Z}(\boldsymbol{W})$ is the set of elements in $\boldsymbol{Z}$ that are not ancestors of $\boldsymbol{W}$ in $\mathcal{G}_{\overline{\boldsymbol{X}}}$.*

Next, we consider operations on equivalence classes starting with a more complete definition of a Maximal Ancestral Graph.

**Definition 9** (Maximal Ancestral Graph). *A mixed graph is ancestral if it does not contain directed or almost directed cycles. It is maximal if, for every pair of nonadjacent vertices $(X, Y)$, there exists a set $\boldsymbol{Z} \subset \boldsymbol{V}$ that $d$-separates them. A Maximal Ancestral Graph (MAG) is a graph that is both ancestral and maximal.*

$$V_1 \overset{\curvearrowleft\curvearrowright}{\phantom{V}} V_2 \leftarrow V_3 \qquad\qquad V_1 \rightarrow V_2 \leftarrow V_3$$

Figure 6: MAG (left), LEG (right).

---

**Algorithm 2** IDP

---

**Input:** A PAG $\mathcal{P}$ and disjoint sets $\boldsymbol{X}, \boldsymbol{Y} \subset \boldsymbol{V}$
**Output:** Expression for $P_{\boldsymbol{x}}(\boldsymbol{y})$ or FAIL
 1: Let $\boldsymbol{D} := \texttt{PossAn}(\boldsymbol{Y})_{\mathcal{P}_{\boldsymbol{V}\setminus\boldsymbol{x}}}$
 2: **return** $\sum_{\boldsymbol{d}\setminus\boldsymbol{y}} \texttt{ID}(\boldsymbol{D}, \boldsymbol{V}, P)$

 3: **function** $\texttt{ID}(\boldsymbol{C}, \boldsymbol{T}, Q = Q[\boldsymbol{T}])$
 4:     if $\boldsymbol{C} = \varnothing$ then return 1.
 5:     if $\boldsymbol{C} = \boldsymbol{T}$ then return $Q$.
     /* In $\mathcal{P}_{\boldsymbol{T}}$, let $\boldsymbol{B}$ denote a bucket, and let $\boldsymbol{C_B}$ denote the $pc$-component of $\boldsymbol{B}$ */
 6:     **if** $\exists \boldsymbol{B} \subset \boldsymbol{T}\setminus\boldsymbol{C}$ such that $\boldsymbol{C_B} \cap \texttt{PossCh}(\boldsymbol{B})_{\mathcal{P}_{\boldsymbol{T}}} \subseteq \boldsymbol{B}$ **then**
 7:         Compute $Q[\boldsymbol{T}\setminus\boldsymbol{B}]$ from $Q[\boldsymbol{T}]$ via (Jaber et al., 2018a, Prop. 2).
 8:         **return** $\texttt{ID}(\boldsymbol{C}, \boldsymbol{T}\setminus\boldsymbol{B}, Q[\boldsymbol{T}\setminus\boldsymbol{B}])$
 9:     **else if** $\exists \boldsymbol{B} \subset \boldsymbol{C}$ such that $\mathcal{R}_{\boldsymbol{B}} \neq \boldsymbol{C}$ **then**
10:         **return** $\texttt{ID}(\mathcal{R}_{\boldsymbol{B}}, \boldsymbol{T}, Q) \times \texttt{ID}(\mathcal{R}_{\boldsymbol{C}\setminus\mathcal{R}_{\boldsymbol{B}}}, \boldsymbol{T}, Q) \,/\, \texttt{ID}(\mathcal{R}_{\boldsymbol{B}} \cap \mathcal{R}_{\boldsymbol{C}\setminus\mathcal{R}_{\boldsymbol{B}}}, \boldsymbol{T}, Q)$
11:     **else**
12:         **return** FAIL
13:     **end if**

---

Given a causal graph over $\boldsymbol{V}$, a unique MAG over $\boldsymbol{V}$ can be constructed such that both independence and non-ancestral relations among $\boldsymbol{V}$ are retained; see e.g. (Zhang, 2008a, Sec. 3). Two MAGs are said to be Markov equivalent if they entail the same set of $d$-separations. Among Markov equivalent MAGs, a particular subset, called Loyal Equivalent Graphs (LEG), can be constructed with the fewest bi-directed edges, all of which are invariant, *i.e.* bi-directed edges in LEGs appear in all MAGs with the same $d$-separations an non-ancestral relations (Zhang and Spirtes, 2005, Corollary 18), and is given in Def. 7. Thus, between a MAG and its LEG, only one kind of difference is possible, namely, some bi-directed edges in the MAG are oriented as directed edges in its LEG, as illustrated in Fig. 6.

An important consequence of the definition of LEGs is that one can traverse the space of Markov equivalent LEGs by checking whether directed edges can be reversed with a simple criterion, restated below from (Zhang and Spirtes, 2012b, Lemma 2).

**Proposition 10** (Transformational characterization of LEGs). *Let $\mathcal{G}$ be a arbitrary LEG, and $X \rightarrow Y$ an arbitrary directed edge in $\mathcal{G}$. The reversal of $X \rightarrow Y$ produces a Markov equivalent LEG if and only if $Pa(X)_{\mathcal{G}} = pa(Y)_{\mathcal{G}}$ and $Sp(X)_{\mathcal{G}} = Sp(Y)_{\mathcal{G}}$.*

*Proof.* The proof can be found in (Zhang and Spirtes, 2012b). $\square$

Two Markov equivalent LEGs can always be transformed to each other via a sequence of reversals according to Prop. 10. Similarly to the definition for PAGs, directed edges $X \rightarrow Y$ in a MAG are said to be visible if there exists no causal graph compatible with this MAG with an edge $X \leftarrow\!-\!-\!-\!-\!\rightarrow Y$, that is unobserved confounding between $X$ and $Y$ can be ruled out. Visibility of an edge can be easily determined by a graphical condition (Zhang, 2008a, Lemma 9). Directed edges that are not visible are called invisible. Prop. 10 is important because, although listing all Markov equivalent MAGs is in general infeasible, one could in principle list all Markov equivalent LEGs by checking reversal of invisible directed edges with this graphical criterion. An explicit algorithm for generating Markov equivalent LEGs is given in Appendix D. Finally, for completeness we reproduce the IDP algorithm (Jaber et al., 2019a) for identifying causal effects from a PAG in Alg. 2.

## A.2 RELATED WORK

We review in this section related work concerned with bounding causal effects with knowledge of fully-specified graph, as no treatment of equivalence classes has been proposed yet.

The natural bounds over the causal effects due to Robins (1989); Manski (1990) were developed with a specific focus on pairs of variables or in studies with imperfect compliance and instrumental variable assumptions. Recently their proof technique have motivated several general works extending these bounds to arbitrary causal diagrams. This was demonstrated recently by (Zhang, 2020; Zhang and Bareinboim, 2019) in which the authors extended earlier bounding strategies to estimate system dynamics in sequential decision-making settings and causal effects in more general graphs. Our work could be interpreted to lie in this line of research, namely extending the natural bounding technique to systems characterized by Partial Ancestral Graphs.

In the partial identification literature, another line of research was pioneered by the seminal work of (Balke and Pearl, 1997) that employs a polynomial optimization program to compute causal bounds and are provably optimal. They proposed a family of canonical models with finite unobserved states, which sufficiently represent all observations and consequences of interventions in instrumental variable models. Based on this canonical characterization, (Balke and Pearl, 1997) reduced the bounding problem to a series of equivalent linear programs. (Chickering and Pearl, 1996) further used Bayesian techniques to investigate the sharpness of these bounds with regard to the observational sample size. Recently, (Zhang et al., 2021; Finkelstein and Shpitser, 2020) describe a polynomial programming approach to solve the partial identification for general causal graphs. They generalize the canonical characterization of SCMs to arbitrary graphs, although require discrete endogenous variables with small support as the time complexity of their algorithm grows exponentially with the size of the support set of variables. In continuous settings, (Gunsilius, 2019) extends the linear programming approach to partial identification of instrumental variable graphs with continuous treatments. Several recent works follow a similar approach: parameterizing causal effects as a linear combinations of a set of fixed basis functions (Padh et al., 2022) or neural networks (Balazadeh Meresht et al., 2022; Hu et al., 2021) and subsequently match the (moments of the) observed distribution while minimizing and maximizing causal effects.

In applications, partial identification has been used in reinforcement learning for the estimation of dynamic treatment regimes (Zhang, 2020; Zhang and Bareinboim, 2019), for estimating policies under safety constraints (Joshi et al., 2024), and within bandit algorithms (Zhang and Bareinboim, 2021; Bellot et al., 2024). And similarly in problems of fairness, for example by Wu et al. (2019).

## A.3 LIMITATIONS

In this work, we start from the assumption that the true PAG that underlies a system of interest can be inferred from data. In general, this requires an assumption of faithfulness, *i.e.* that the independencies in data imply a corresponding separation in the underlying causal diagram, and an oracle for testing for conditional independencies. Learning the true PAG from finite data can be a significant challenge in practice (Spirtes et al., 2000; Robins et al., 2003; Zhang and Spirtes, 2012a; Bellot et al., 2022; Bellot and van der Schaar, 2021). In higher-dimensional systems, the computational complexity of estimating the conditional distributions that define lower and upper bounds on causal effects is another substantial challenge. In light of this, it is important to make the distinction between the task of partial identification, that is inferring an expression to bound causal effects, and that of causal effect estimation, that is providing efficient estimators from finite samples to compute bounds in practice. This set of results is concerned with the first task (partial identification). The objective of our procedure is to decide whether the effect can be bounded and provide an expression for lower and upper bounds, while being agnostic as to whether $P(V)$ can be accurately estimated from the available samples. Several works consider the efficient estimation of identifiable causal effects (Jung et al., 2021). Extending these techniques to the problem of bounding non-identifiable causal effects is an important direction for future work. Finally, we emphasize that simulations on real and synthetic data are provided for illustration purposes only. These results do not recommend or advocate for the implementation of a particular intervention, and should be considered in practice in combination with other aspects of the decision-making process.

# B PROOFS

**Prop. 1 restated**. *Given a PAG $\mathcal{P}$ over $\boldsymbol{V}$ and $\boldsymbol{A} \subset \boldsymbol{C} \subseteq \boldsymbol{V}$, let the region of $\boldsymbol{A}$ with respect to $\boldsymbol{C}$ be denoted $\mathcal{R}_{\boldsymbol{A}}$. $Q[\boldsymbol{C}]$ can be decomposed as,*

$$Q[\boldsymbol{C}] = Q[\mathcal{R}_{\boldsymbol{A}}] \cdot Q[\mathcal{R}_{\boldsymbol{C} \setminus \boldsymbol{A}}] \,/\, Q[\mathcal{R}_{\boldsymbol{A}} \cap \mathcal{R}_{\boldsymbol{C} \setminus \boldsymbol{A}}]. \tag{20}$$

*Proof.* The proof can be found in (Jaber et al., 2019a, Thm.1). $\qquad\square$

**Prop. 2 restated**. *Let $\mathcal{P}$ be the PAG underlying $P(\boldsymbol{V})$. Under faithfulness, a causal effect is partially identifiable from $P(\boldsymbol{V})$ with bound $[a, b]$ if and only if it is partially identifiable from $\mathcal{P}$ and $P(\boldsymbol{V})$ with bound $[a, b]$.*

*Proof.* Let $\mathcal{P}$ be the PAG underlying a given distribution $P(\boldsymbol{V})$. Let $\mathbb{M}$ be the set of all SCMs over a set of endogenous variables $\boldsymbol{V}$, and let $\mathbb{M}(\mathcal{P})$ be the subset of SCMs over a set of endogenous variables $\boldsymbol{V}$ whose induced causal diagrams are consistent with the PAG $\mathcal{P}$. The partial identification problem from $P(\boldsymbol{V})$ (Def. 2) may be stated as follows,

$$\min/\max_{\mathcal{M} \in \mathbb{M}} P_{\mathcal{M}}(\boldsymbol{y_x}), \quad \text{such that} \quad P_{\mathcal{M}}(\boldsymbol{v}) = P(\boldsymbol{v}). \tag{21}$$

Similarly, the partial identification problem from the PAG $\mathcal{P}$ and $P(\boldsymbol{V})$ (Def. 6) may be stated as follows,

$$\min/\max_{\mathcal{M} \in \mathbb{M}(\mathcal{P})} P_{\mathcal{M}}(\boldsymbol{y_x}), \quad \text{such that} \quad P_{\mathcal{M}}(\boldsymbol{v}) = P(\boldsymbol{v}). \tag{22}$$

These definitions highlight that the bounding problem involves a search over structural models consistent with the observational distribution $P(\boldsymbol{V})$, and optionally $\mathcal{P}$. We will show that under faithfulness, $\{\mathcal{M} \in \mathbb{M} : P_{\mathcal{M}}(\boldsymbol{v}) = P(\boldsymbol{v})\} = \{\mathcal{M} \in \mathbb{M}(\mathcal{P}) : P_{\mathcal{M}}(\boldsymbol{v}) = P(\boldsymbol{v})\}$. In that case, the optimization problems coincide and therefore their solutions coincide.

To see this note that $\{\mathcal{M} \in \mathbb{M}(\mathcal{P}) : P_{\mathcal{M}}(\boldsymbol{v}) = P(\boldsymbol{v})\} \subseteq \{\mathcal{M} \in \mathbb{M} : P_{\mathcal{M}}(\boldsymbol{v}) = P(\boldsymbol{v})\}$ by definition since $\mathbb{M}(\mathcal{P})$ introduces a restriction on the space $\mathbb{M}$. Under faithfulness, consider any SCM $\mathcal{M} \in \mathbb{M}$ such that $P_{\mathcal{M}}(\boldsymbol{v}) = P(\boldsymbol{v})$. It follows then that,

$$(\boldsymbol{X} \perp\!\!\!\perp \boldsymbol{Y} \mid \boldsymbol{Z})_{P_{\mathcal{M}}} \Leftrightarrow (\boldsymbol{X} \perp\!\!\!\perp \boldsymbol{Y} \mid \boldsymbol{Z})_{\mathcal{G}_{\mathcal{M}}}.$$

In other words, under faithfulness, the graph $\mathcal{G}_{\mathcal{M}}$ entails a $d$-separation for every conditional independence in the data, and vice versa. $\mathcal{G}_{\mathcal{M}}$ must then be included in the set of diagrams represented by the PAG $\mathcal{P}$ as the PAG is defined as the set of diagrams with $d$-separation statements match the conditional independencies in data, that is $\mathcal{M} \in \mathbb{M}(\mathcal{P})$. As $\mathcal{M}$ was arbitrary (up to agreement with the observational distribution), we have that, under faithfulness, $\{\mathcal{M} \in \mathbb{M} : P_{\mathcal{M}}(\boldsymbol{v}) = P(\boldsymbol{v})\} \subseteq \{\mathcal{M} \in \mathbb{M}(\mathcal{P}) : P_{\mathcal{M}}(\boldsymbol{v}) = P(\boldsymbol{v})\}$. This implies then that $\{\mathcal{M} \in \mathbb{M} : P_{\mathcal{M}}(\boldsymbol{v}) = P(\boldsymbol{v})\} = \{\mathcal{M} \in \mathbb{M}(\mathcal{P}) : P_{\mathcal{M}}(\boldsymbol{v}) = P(\boldsymbol{v})\}$ showing the claim. $\qquad\square$

Next we introduce two utility lemmas that will be useful in the derivation of the following proofs.

**Lemma 1.** *Given a PAG $\mathcal{P}$ over $\boldsymbol{V}$, let $\boldsymbol{C} \subset \boldsymbol{V}$. Then,*

$$\left\{ Q[\boldsymbol{C}; \mathcal{M}] : \mathcal{M} \in \mathbb{M}(\mathcal{P}) \right\} = \left\{ Q[\boldsymbol{C}; \mathcal{M}] : \mathcal{M} \in \mathbb{M}(\mathcal{P}_{\widetilde{\boldsymbol{V} \setminus \boldsymbol{C}}}) \right\}. \tag{23}$$

*Proof.* For a given causal diagram $\mathcal{G}$, a causal effect of interest $P_{\boldsymbol{x}}(\boldsymbol{y})$ can be written,

$$P_{\boldsymbol{x}}(\boldsymbol{y}) = \sum_{\boldsymbol{v} \setminus \{\boldsymbol{x} \cup \boldsymbol{y}\}} \int_{\Omega_U} \prod_{V \in \boldsymbol{V} \setminus \boldsymbol{X}} P(v \mid \boldsymbol{pa}_V, \boldsymbol{u}_V) dP(\boldsymbol{u}) = \sum_{\boldsymbol{s} \setminus \{\boldsymbol{y}\}} \int_{\Omega_U} \prod_{V \in \boldsymbol{S}} P(v \mid \boldsymbol{pa}_V, \boldsymbol{u}_V) dP(\boldsymbol{u}),$$

where $\boldsymbol{S} = An(\boldsymbol{Y})_{\mathcal{G}_{\overline{\boldsymbol{X}}}} \setminus \boldsymbol{X}$ and $\boldsymbol{s}$ is some value in the domain of $\boldsymbol{S}$. This expression depends only on probabilities associated with variables $\boldsymbol{S}, \boldsymbol{U}_S, S \in \boldsymbol{S}$ and their functional dependencies through terms $P(v \mid \boldsymbol{pa}_V, \boldsymbol{u}_V)$ that in turn are determined by the underlying SCMs compatible with the graph and data, which in this case are parameterized by $\{f_V : V \in \boldsymbol{S}\}$ and $\{P(u_V) : V \in An(\boldsymbol{S})_{\mathcal{G}[\boldsymbol{C}]}\}$ where $\boldsymbol{C}$ is the $c$-component in $\mathcal{G}$ that contains $\boldsymbol{S}$. The distribution and values of any other variable is of no consequence to the desired causal effect. In particular, any descendants of $\boldsymbol{Y}$ in $\mathcal{G}$ can be marginalized out without loss of generality.

The same reasoning applies to $Q[\boldsymbol{C}] := P_{\boldsymbol{v} \backslash \boldsymbol{c}}(\boldsymbol{c})$ which depends only on $An(\boldsymbol{C})_{\mathcal{G}_{\overline{\boldsymbol{V} \backslash \boldsymbol{C}}}} \backslash (\boldsymbol{V} \backslash \boldsymbol{C})$ and the functional dependencies of associated variables. Recall that $\mathcal{P}_{\overline{\boldsymbol{V} \backslash \boldsymbol{C}}}$ denotes the graph in which all edges that are visible in $\mathcal{P}$ and are into variables in $\boldsymbol{V} \backslash \boldsymbol{C}$ are deleted, and in which all invisible edges that are into variables in $\boldsymbol{V} \backslash \boldsymbol{C}$ are replaced with bi-directed edges. Since $\mathcal{P}_{\overline{\boldsymbol{V} \backslash \boldsymbol{C}}}$ only modifies the functional assignment of descendants of $\boldsymbol{C}$ and these do not influence the causal effect of interest we have that,

$$\left\{ Q[\boldsymbol{C}; \mathcal{M}] : \mathcal{M} \in \mathbb{M}(\mathcal{P}) \right\} = \left\{ Q[\boldsymbol{C}; \mathcal{M}] : \mathcal{M} \in \mathbb{M}(\mathcal{P}_{\overline{\boldsymbol{V} \backslash \boldsymbol{C}}}) \right\}. \tag{24}$$

$\square$

This proposition implies that it is sufficient to consider the set of causal diagrams compatible with $\mathcal{P}_{\overline{\boldsymbol{V} \backslash \boldsymbol{C}}}$, i.e. the set of ME causal diagrams $\mathcal{G}$ in which edges $X \to Y, X \in pa(\boldsymbol{C})_{\mathcal{G}}, Y \in \boldsymbol{C}$ have been removed, to characterize lower and upper bounds over queries of the form $Q[\boldsymbol{C}]$ from $\mathcal{P}$. The next lemma shows how to compute quantities $Q[\cdot]$ from larger ones and is an extension of an analogous results defined for causal diagrams Tian and Pearl (2002).

**Lemma 2.** *Let $\boldsymbol{W} \subseteq \boldsymbol{C} \subset \boldsymbol{V}$ such that $\boldsymbol{W} = \texttt{PossAn}(\boldsymbol{W})_{\mathcal{P}_{\boldsymbol{C}}}$. Then, $Q[\boldsymbol{W}] = \sum_{\boldsymbol{c} \backslash \boldsymbol{w}} Q[\boldsymbol{C}]$.*

*Proof.* By (Jaber et al., 2018a, Prop. 1), if $X$ is an ancestor of $Y$ in a causal diagram $\mathcal{G}_{\boldsymbol{C}}$, then $X$ is a possible ancestor of $Y$ in the PAG $\mathcal{P}_{\boldsymbol{C}}$. By the converse of this implication, if $X$ is not a possible ancestor of $Y$ $\mathcal{P}_{\boldsymbol{C}}$, then $X$ is not an ancestor of $Y$ in $\mathcal{G}_{\boldsymbol{C}}$. Let $\boldsymbol{W} \subseteq \boldsymbol{C} \subset \boldsymbol{V}$ such that $\boldsymbol{W} = \texttt{PossAn}(\boldsymbol{W})_{\mathcal{P}_{\boldsymbol{C}}}$. By (Jaber et al., 2018a, Prop. 1) therefore, no variable in $\boldsymbol{R} = \boldsymbol{C} \backslash \boldsymbol{W}$ being a possible ancestor of $\boldsymbol{W}$ in $\mathcal{P}_{\boldsymbol{C}}$ implies that no variable in $\boldsymbol{R} = \boldsymbol{C} \backslash \boldsymbol{W}$ is an ancestor of $\boldsymbol{W}$ in any ME causal diagram $\mathcal{G}_{\boldsymbol{C}}$. For any causal diagram $\mathcal{G}_{\boldsymbol{C}}$, by (Tian and Pearl, 2003, Lemma 3) $Q[\boldsymbol{W}] = \sum_{\boldsymbol{r}} Q[\boldsymbol{C}]$. Moreover, if $Q[\boldsymbol{C}]$ could be uniquely computed from $P(\boldsymbol{V})$ and $\mathcal{P}$, the $Q[\boldsymbol{W}]$ could be uniquely computed from $P(\boldsymbol{V})$ and $\mathcal{P}$. $\square$

We are now ready to prove the validity of lower and upper bounds.

**Prop. 3 restated**. *Given a PAG $\mathcal{P}$, consider sets $\boldsymbol{S} \subset \boldsymbol{C} \subseteq \boldsymbol{V}$ and define $\boldsymbol{W} = \texttt{PossAn}(\boldsymbol{S})_{\mathcal{P}_{\boldsymbol{C}}}$, $\boldsymbol{R} = \boldsymbol{W} \backslash \boldsymbol{S}$, and $\boldsymbol{T} = \texttt{PossSp}(\boldsymbol{S})_{\mathcal{P}_{\boldsymbol{C}}} \backslash \boldsymbol{S}$. Let $\boldsymbol{A}, \boldsymbol{B}$ partition $\boldsymbol{R}$ such that $\boldsymbol{B} = \texttt{PossDe}(\boldsymbol{T})_{\mathcal{P}_{\boldsymbol{C}}} \cap \boldsymbol{R}, \boldsymbol{A} = \boldsymbol{R} \backslash \boldsymbol{B}$. $Q[\boldsymbol{S}]$ is lower bounded as follows:*

$$Q[\boldsymbol{S}] \geqslant \max_{\boldsymbol{z}} \frac{Q[\boldsymbol{W}]}{\sum_{\boldsymbol{s}, \boldsymbol{b}} Q[\boldsymbol{W}]}, \tag{25}$$

*where $\boldsymbol{Z} = \texttt{PossPa}(\boldsymbol{W})_{\mathcal{P}} \backslash \texttt{PossPa}(\boldsymbol{S})_{\mathcal{P}}$.*

*Proof.* If $Q[\boldsymbol{S}]$ is not identifiable given $\mathcal{P}$, then $Q[\boldsymbol{S}]$ is not identifiable in one or more ME causal diagrams $\mathcal{G}$ by (Jaber et al., 2019a, Thm. 4). In each of those diagrams $\mathcal{G}$, then there must exist an open backdoor path from a node in $\boldsymbol{V} \backslash \boldsymbol{S}$ to a node in $\boldsymbol{S}$ that could be blocked with access to a set of unobserved confounders $\boldsymbol{U}$. Let $\boldsymbol{U}$ be the union of exogenous variables that block such open backdoor paths. In turn, let $\boldsymbol{S} \subset \boldsymbol{C}$, where $\boldsymbol{C}$ is a $pc$-component in a sub-graph of $\mathcal{P}$ with $Q[\boldsymbol{C}]$ identifiable. Following the statement of the proposition, let $\boldsymbol{W} = \texttt{PossAn}(\boldsymbol{S})_{\mathcal{P}_{\boldsymbol{C}}}$, $\boldsymbol{R} = \boldsymbol{W} \backslash \boldsymbol{S}$, and $\boldsymbol{T} = \texttt{PossSp}(\boldsymbol{S})_{\mathcal{P}_{\boldsymbol{C}}} \backslash \boldsymbol{S}$. Further, let $\boldsymbol{A}, \boldsymbol{B}$ partition $\boldsymbol{R}$ such that $\boldsymbol{B} = \texttt{PossDe}(\boldsymbol{T})_{\mathcal{P}_{\boldsymbol{C}}}, \boldsymbol{A} = \boldsymbol{R} \backslash \boldsymbol{B}$. Without loss of generality, by Lem. 1 inference on $Q[\boldsymbol{S}]$ given $\mathcal{P}$ is equivalent to inference on $Q[\boldsymbol{S}]$ given $\mathcal{P}_{\overline{\boldsymbol{V} \backslash \boldsymbol{S}}}$.

To show the claim, we show that $Q[\boldsymbol{S}]$ is lower bounded in every causal diagram compatible with $\mathcal{P}_{\overline{\boldsymbol{V} \backslash \boldsymbol{S}}}$. Let $\mathcal{G}$ be any such causal diagram. In light of the definitions above, it holds that,

1. $\boldsymbol{R}$ is $d$-separated from $\boldsymbol{S}$ conditioned on $\boldsymbol{U}$ in $\mathcal{G}_{\overline{\boldsymbol{V} \backslash \boldsymbol{W}}, \underline{\boldsymbol{R}}}$,

2. $\boldsymbol{U}$ is exogenous and thus $d$-separated from $\boldsymbol{R}$ conditioned on $\boldsymbol{V} \backslash \boldsymbol{W}$ in $\mathcal{G}_{\overline{\boldsymbol{V} \backslash \boldsymbol{W}}, \boldsymbol{R}}$,

3. $\boldsymbol{A}$ is $d$-separated from $\boldsymbol{U}$ conditioned on $\boldsymbol{V} \backslash \boldsymbol{W}$ in $\mathcal{G}_{\overline{\boldsymbol{V} \backslash \boldsymbol{W}}}$.

Condition (1) states that all backdoor paths from $\boldsymbol{R}$ to $\boldsymbol{S}$, *i.e.* those starting with an edge $R \leftarrow \cdots$ or $R \leftarrow\!-\!-\!-\!\rightarrow \cdots$, $R \in \boldsymbol{R}$, are blocked conditioned on $\boldsymbol{U}$. To see this, note that $\boldsymbol{U}$ is chosen to block all backdoor paths through an unobserved confounder and that any other backdoor paths are assumed away in $\mathcal{P}_{\overline{\boldsymbol{V} \backslash \boldsymbol{S}}}$, *i.e.* there is no edge of the form $X \to Y, X \in \boldsymbol{S}, Y \in \boldsymbol{V} \backslash \boldsymbol{S}$ in any causal diagram compatible with $\mathcal{P}_{\overline{\boldsymbol{V} \backslash \boldsymbol{S}}}$. Condition (2) holds because $\boldsymbol{U}$ is exogenous; in $\mathcal{G}_{\overline{\boldsymbol{V} \backslash \boldsymbol{W}}, \boldsymbol{R}}$ no open path between $\boldsymbol{R}$ and $\boldsymbol{U}$ conditioned on $\boldsymbol{V} \backslash \boldsymbol{W}$ could exist. Condition (3) holds because $\boldsymbol{A}$ is defined precisely as the

set of variables in $\boldsymbol{R}$ that are not descendants of $\boldsymbol{T}$ and thus that are not descendants of $\boldsymbol{U}$; any path between $\boldsymbol{A}$ and $\boldsymbol{U}$ is therefore blocked by a child or descendant of $\boldsymbol{U}$ that acts as a collider on the path.

The following derivation applies to $Q[\boldsymbol{S}]$ in $\mathcal{G}$.

$$
\begin{aligned}
Q[\boldsymbol{S}] &:= P(\boldsymbol{s} \mid do(\boldsymbol{v}\backslash\boldsymbol{s})) \\
&= P(\boldsymbol{s} \mid do(\boldsymbol{r}, \boldsymbol{v}\backslash\boldsymbol{w})) \\
&\overset{(1)}{=} \sum_{\boldsymbol{u}} P(\boldsymbol{s} \mid \boldsymbol{u}, \boldsymbol{r}, do(\boldsymbol{v}\backslash\boldsymbol{w}))P(\boldsymbol{u} \mid do(\boldsymbol{r}, \boldsymbol{v}\backslash\boldsymbol{w})) \\
&\overset{(2)}{=} \sum_{\boldsymbol{u}} P(\boldsymbol{s} \mid \boldsymbol{u}, \boldsymbol{r}, do(\boldsymbol{v}\backslash\boldsymbol{w}))P(\boldsymbol{u} \mid do(\boldsymbol{v}\backslash\boldsymbol{w})) \\
&\overset{(3)}{=} \sum_{\boldsymbol{u}} P(\boldsymbol{s} \mid \boldsymbol{u}, \boldsymbol{a}, \boldsymbol{b}, do(\boldsymbol{v}\backslash\boldsymbol{w}))P(\boldsymbol{u} \mid \boldsymbol{a}, do(\boldsymbol{v}\backslash\boldsymbol{w})) \\
&\geqslant \sum_{\boldsymbol{u}} P(\boldsymbol{s} \mid \boldsymbol{u}, \boldsymbol{a}, \boldsymbol{b}, do(\boldsymbol{v}\backslash\boldsymbol{w}))P(\boldsymbol{u}, \boldsymbol{b} \mid \boldsymbol{a}, do(\boldsymbol{v}\backslash\boldsymbol{w})) \\
&= P(\boldsymbol{s}, \boldsymbol{b} \mid \boldsymbol{a}, do(\boldsymbol{v}\backslash\boldsymbol{w})) \\
&= \frac{P(\boldsymbol{w} \mid do(\boldsymbol{v}\backslash\boldsymbol{w}))}{\sum_{\boldsymbol{s}, \boldsymbol{b}} P(\boldsymbol{w} \mid do(\boldsymbol{v}\backslash\boldsymbol{w}))} \\
&= \frac{Q[\boldsymbol{W}]}{\sum_{\boldsymbol{s}, \boldsymbol{b}} Q[\boldsymbol{W}]}.
\end{aligned}
$$

(1) follows from condition 1; (2) follows from condition 2; (3) follows from condition 3. The inequality follows from the fact that the event $\{\boldsymbol{u}, \boldsymbol{b}\}$ is less likely than event $\{\boldsymbol{u}\}$ under any probability mass function. Finally, $Q[\boldsymbol{S}]$ is a function of $Pa(\boldsymbol{S})_{\mathcal{G}}$ only and thus this bound holds for any value of $\boldsymbol{Z} := Pa(\boldsymbol{W})_{\mathcal{G}}\backslash Pa(\boldsymbol{S})_{\mathcal{G}}$ and therefore any $\boldsymbol{Z} = \texttt{PossPa}(\boldsymbol{W})_{\mathcal{P}}\backslash\texttt{PossPa}(\boldsymbol{S})_{\mathcal{P}}$. In particular,

$$
Q[\boldsymbol{S}] \geqslant \max_{\boldsymbol{z}} \frac{Q[\boldsymbol{W}]}{\sum_{\boldsymbol{s}, \boldsymbol{b}} Q[\boldsymbol{W}]}.
$$

$\square$

**Prop. 4 restated**. *Given a PAG $\mathcal{P}$, consider sets $\boldsymbol{S} \subset \boldsymbol{C} \subseteq \boldsymbol{V}$ and let a partial topological ordering $\boldsymbol{S}$ be $\boldsymbol{S}_1 < \cdots < \boldsymbol{S}_k$. Define $\boldsymbol{W} = \texttt{PossAn}(\boldsymbol{S})_{\mathcal{P}_C}$, $\boldsymbol{R} = \boldsymbol{W}\backslash\boldsymbol{S}$, $\boldsymbol{T} = \texttt{PossSp}(\boldsymbol{S})_{\mathcal{P}_C}\backslash\boldsymbol{S}$, and $\boldsymbol{T} = \texttt{PossSp}(\boldsymbol{S})_{\mathcal{P}_C}\backslash\boldsymbol{S}$. Let $\boldsymbol{A}, \boldsymbol{B}$ partition $\boldsymbol{R}$ such that $\boldsymbol{B} = \texttt{PossDe}(\boldsymbol{T})_{\mathcal{P}_C} \cap \boldsymbol{R}, \boldsymbol{A} = \boldsymbol{R}\backslash\boldsymbol{B}$. $Q[\boldsymbol{S}]$ is upper bounded as follows:*

$$
Q[\boldsymbol{S}] \leqslant \min_{\boldsymbol{z}} \left\{ \frac{Q[\boldsymbol{W}]}{\sum_{\boldsymbol{s}, \boldsymbol{b}} Q[\boldsymbol{W}]} - \sum_{\boldsymbol{s}_k} \frac{Q[\boldsymbol{W}]}{\sum_{\boldsymbol{s}, \boldsymbol{b}} Q[\boldsymbol{W}]} \right\} + Q[\boldsymbol{S}\backslash\boldsymbol{S}_k], \tag{26}
$$

*where $\boldsymbol{Z} = \texttt{PossPa}(\boldsymbol{W})_{\mathcal{P}}\backslash\texttt{PossPa}(\boldsymbol{S})_{\mathcal{P}}$.*

*Proof.* Similarly to the proof of Prop. 3, to show the claim we show that $Q[\boldsymbol{S}]$ is upper bounded in every causal diagram compatible with $\tilde{\mathcal{P}}$ and rely on facts (1,2,3) above. Let $\mathcal{G}$ be any such causal diagram. Let a partial topological ordering of $\boldsymbol{S}$ be $\boldsymbol{S}_1 < \cdots < \boldsymbol{S}_k$.

It then holds that,

$$
\begin{aligned}
Q[\boldsymbol{S}] :=\ & P(\boldsymbol{s} \mid do(\boldsymbol{v}\backslash\boldsymbol{s})) \\
=\ & P(\boldsymbol{s} \mid do(\boldsymbol{r}, \boldsymbol{v}\backslash\boldsymbol{w})) \\
=\ & \sum_{\boldsymbol{u}} P(\boldsymbol{s} \mid \boldsymbol{u}, \boldsymbol{r}, do(\boldsymbol{v}\backslash\boldsymbol{w})) P(\boldsymbol{u} \mid do(\boldsymbol{r}, \boldsymbol{v}\backslash\boldsymbol{w})) \\
=\ & \sum_{\boldsymbol{u}} P(\boldsymbol{s} \mid \boldsymbol{u}, \boldsymbol{r}, do(\boldsymbol{v}\backslash\boldsymbol{w})) P(\boldsymbol{u} \mid do(\boldsymbol{v}\backslash\boldsymbol{w})) \\
=\ & \sum_{\boldsymbol{u}} P(\boldsymbol{s} \mid \boldsymbol{u}, \boldsymbol{a}, \boldsymbol{b}, do(\boldsymbol{v}\backslash\boldsymbol{w})) P(\boldsymbol{u} \mid \boldsymbol{a}, do(\boldsymbol{v}\backslash\boldsymbol{w})) \\
=\ & \sum_{\boldsymbol{u}} P(\boldsymbol{s} \mid \boldsymbol{u}, \boldsymbol{a}, \boldsymbol{b}, do(\boldsymbol{v}\backslash\boldsymbol{w})) \Big( P(\boldsymbol{u}, \boldsymbol{b} \mid \boldsymbol{a}, do(\boldsymbol{v}\backslash\boldsymbol{w})) + P(\boldsymbol{u} \mid \boldsymbol{a}, do(\boldsymbol{v}\backslash\boldsymbol{w})) - P(\boldsymbol{u}, \boldsymbol{b} \mid \boldsymbol{a}, do(\boldsymbol{v}\backslash\boldsymbol{w})) \Big) \\
\overset{(1)}{=}\ & \frac{Q[\boldsymbol{W}]}{\sum_{\boldsymbol{s},\boldsymbol{b}} Q[\boldsymbol{W}]} + \sum_{\boldsymbol{u}} P(\boldsymbol{s} \mid \boldsymbol{u}, \boldsymbol{a}, \boldsymbol{b}, do(\boldsymbol{v}\backslash\boldsymbol{w})) \Big( P(\boldsymbol{u} \mid \boldsymbol{a}, do(\boldsymbol{v}\backslash\boldsymbol{w})) - P(\boldsymbol{u}, \boldsymbol{b} \mid \boldsymbol{a}, do(\boldsymbol{v}\backslash\boldsymbol{w})) \Big) \\
\overset{(2)}{\leqslant}\ & \frac{Q[\boldsymbol{W}]}{\sum_{\boldsymbol{s},\boldsymbol{b}} Q[\boldsymbol{W}]} + \sum_{\boldsymbol{u}} P(\boldsymbol{s}\backslash\boldsymbol{s}_k \mid \boldsymbol{u}, \boldsymbol{a}, \boldsymbol{b}, do(\boldsymbol{v}\backslash\boldsymbol{w})) \Big( P(\boldsymbol{u} \mid \boldsymbol{a}, do(\boldsymbol{v}\backslash\boldsymbol{w})) - P(\boldsymbol{u}, \boldsymbol{b} \mid \boldsymbol{a}, do(\boldsymbol{v}\backslash\boldsymbol{w})) \Big) \\
\overset{(3)}{=}\ & \frac{Q[\boldsymbol{W}]}{\sum_{\boldsymbol{s},\boldsymbol{b}} Q[\boldsymbol{W}]} + \sum_{\boldsymbol{u}} P(\boldsymbol{s}\backslash\boldsymbol{s}_k \mid \boldsymbol{u}, \boldsymbol{a}, \boldsymbol{b}, do(\boldsymbol{v}\backslash\boldsymbol{w}, \boldsymbol{s}_k)) P(\boldsymbol{u} \mid \boldsymbol{a}, do(\boldsymbol{v}\backslash\boldsymbol{w}, \boldsymbol{s}_k)) \\
& - \sum_{\boldsymbol{u},\boldsymbol{s}_k} P(\boldsymbol{s} \mid \boldsymbol{u}, \boldsymbol{a}, \boldsymbol{b}, do(\boldsymbol{v}\backslash\boldsymbol{w})) P(\boldsymbol{u}, \boldsymbol{b} \mid \boldsymbol{a}, do(\boldsymbol{v}\backslash\boldsymbol{w})) \\
\overset{(4)}{=}\ & \frac{Q[\boldsymbol{W}]}{\sum_{\boldsymbol{s},\boldsymbol{b}} Q[\boldsymbol{W}]} + P(\boldsymbol{s}\backslash\boldsymbol{s}_k \mid \boldsymbol{a}, do(\boldsymbol{v}\backslash\boldsymbol{w}, \boldsymbol{b}, \boldsymbol{s}_k)) - \sum_{\boldsymbol{s}_k} \frac{Q[\boldsymbol{W}]}{\sum_{\boldsymbol{s},\boldsymbol{b}} Q[\boldsymbol{W}]} \\
\overset{(5)}{=}\ & \frac{Q[\boldsymbol{W}]}{\sum_{\boldsymbol{s},\boldsymbol{b}} Q[\boldsymbol{W}]} + P(\boldsymbol{s}\backslash\boldsymbol{s}_k \mid do(\boldsymbol{v}\backslash\boldsymbol{w}, \boldsymbol{a}, \boldsymbol{b}, \boldsymbol{s}_k)) - \sum_{\boldsymbol{s}_k} \frac{Q[\boldsymbol{W}]}{\sum_{\boldsymbol{s},\boldsymbol{b}} Q[\boldsymbol{W}]} \\
\overset{(6)}{=}\ & \frac{Q[\boldsymbol{W}]}{\sum_{\boldsymbol{s},\boldsymbol{b}} Q[\boldsymbol{W}]} + Q[\boldsymbol{S}\backslash\boldsymbol{S}_k] - \sum_{\boldsymbol{s}_k} \frac{Q[\boldsymbol{W}]}{\sum_{\boldsymbol{s},\boldsymbol{b}} Q[\boldsymbol{W}]}.
\end{aligned}
$$

The first four equalities follows from the observations (1,2,3) as in the derivation of the lower bound (Prop. 3); (1) follows from the derivation in the lower bound; (2) follows from the fact that the event $\{\boldsymbol{s}\}$ is less likely than event $\{\boldsymbol{s}\backslash\boldsymbol{s}_k\}$ under any probability mass function and that the difference in brackets is greater or equal to zero; (3) follows by rule 3 of the do-calculus (Thm. 1) since $\boldsymbol{S}_k$ is $d$-separated from $\boldsymbol{S}\backslash\boldsymbol{S}_k$ given $\boldsymbol{U}, \boldsymbol{A}, \boldsymbol{B}, \boldsymbol{V}\backslash\boldsymbol{W}$ in $\mathcal{G}_{\overline{\boldsymbol{S}_k, \boldsymbol{V}\backslash\boldsymbol{W}}}$ and since $\boldsymbol{S}_k$ is $d$-separated from $\boldsymbol{U}$ given $\boldsymbol{A}$ in $\mathcal{G}_{\overline{\boldsymbol{S}_k, \boldsymbol{V}\backslash\boldsymbol{W}}}$; (4) follows by marginalizing out $\boldsymbol{U}$ and similarly to (1); (5) follows by the rule 2 of do-calculus (Thm. 1) since $\boldsymbol{S}\backslash\boldsymbol{S}_k$ is $d$-separated from $\boldsymbol{A}$ in $\mathcal{G}_{\overline{\boldsymbol{V}\backslash\boldsymbol{W}, \boldsymbol{B}, \boldsymbol{S}_k}\underline{\boldsymbol{A}}}$; (6) follows by the definition of $Q[\cdot]$.

Similarly $P(\boldsymbol{s} \mid do(\boldsymbol{v}\backslash\boldsymbol{s}))$ is a function of $Pa(\boldsymbol{S})_{\mathcal{G}}$ only and thus this bound holds for any value of $\boldsymbol{Z} := Pa(\boldsymbol{W})_{\mathcal{G}}\backslash Pa(\boldsymbol{S})_{\mathcal{G}}$. In particular,

$$
Q[\boldsymbol{S}] \leqslant \min_{\boldsymbol{z}} \left\{ \frac{Q[\boldsymbol{W}]}{\sum_{\boldsymbol{s},\boldsymbol{b}} Q[\boldsymbol{W}]} - \sum_{\boldsymbol{s}_k} \frac{Q[\boldsymbol{W}]}{\sum_{\boldsymbol{s},\boldsymbol{b}} Q[\boldsymbol{W}]} \right\} + Q[\boldsymbol{S}\backslash\boldsymbol{S}_k].
$$

$\square$

**Prop. 5 restated.** *Consider a query $P_{\boldsymbol{x}}(\boldsymbol{y})$ and let $\mathcal{P}$ be the PAG over $\{\boldsymbol{X}, \boldsymbol{Y}\}$ compatible with P. Then, under an assumption of faithfulness, the bounds given in Props. 3 and 4 are at least as tight as the natural bounds.*

*Proof.* Following the premise of the proposition, consider a query $P_{\boldsymbol{x}}(\boldsymbol{y})$ and let $\mathcal{P}$ be the PAG over $\boldsymbol{V} = \{\boldsymbol{X}, \boldsymbol{Y}\}$ compatible with P. $P_{\boldsymbol{x}}(\boldsymbol{y}) = Q[\boldsymbol{Y}]$ and therefore we will compare bounds on $Q[\boldsymbol{Y}]$ with the natural bounds.

By Lem. 2, for any $\boldsymbol{W} = \texttt{PossAn}(\boldsymbol{S})_{\mathcal{P}}$ it holds that $Q[\boldsymbol{W}] = \sum_{\boldsymbol{v}\backslash\boldsymbol{w}} Q[\boldsymbol{V}] = \sum_{\boldsymbol{v}\backslash\boldsymbol{w}} P(\boldsymbol{v}) = P(\boldsymbol{w})$. Further, since $\boldsymbol{W} \subseteq \{\boldsymbol{X}, \boldsymbol{Y}\}$ it holds that $P(\boldsymbol{w}) \geqslant P(\boldsymbol{x}, \boldsymbol{y})$. The proposed lower bound can then be shown to be larger or equal to the

natural lower bound as

$$\max_{\boldsymbol{z}} \frac{Q[\boldsymbol{W}]}{\sum_{\boldsymbol{s},\boldsymbol{b}} Q[\boldsymbol{W}]} \geqslant \frac{Q[\boldsymbol{W}]}{\sum_{\boldsymbol{s},\boldsymbol{b}} Q[\boldsymbol{W}]} \geqslant Q[\boldsymbol{W}] \geqslant P(\boldsymbol{x},\boldsymbol{y}).$$

The last term being the expression of the natural lower bound.

Let $\boldsymbol{S} = \boldsymbol{Y}, \boldsymbol{R} = \boldsymbol{W}\backslash\boldsymbol{S}$, and let $\boldsymbol{S}_k \subseteq \boldsymbol{S}$ be any subset of $\boldsymbol{S}$. Note that $\boldsymbol{R} \cup (\boldsymbol{V}\backslash\boldsymbol{W}) = \boldsymbol{X}$ in this example. From the definition of $\boldsymbol{W}$ as the set of possible ancestors of $\boldsymbol{Y}$ in $\mathcal{P}$, it holds that $(\boldsymbol{R} \perp\!\!\!\perp \boldsymbol{V}\backslash\boldsymbol{W})_{\mathcal{G}_{\overline{\boldsymbol{V}\backslash\boldsymbol{W}}}}$ for any causal diagram $\mathcal{G}$ compatible with an arbitrary $\mathcal{P}$ as there cannot be any directed paths from $\boldsymbol{V}\backslash\boldsymbol{W}$ to $\boldsymbol{R}$ (otherwise at least some element in $\boldsymbol{V}\backslash\boldsymbol{W}$ would be defined as a possible ancestor of $\boldsymbol{S}$ which we ruled out by the definition of $\boldsymbol{W}$).

Denote $\boldsymbol{U}$ the set of unobserved confounders. It holds then that conditioning on $\boldsymbol{U}$ blocks all backdoor paths from $\boldsymbol{R}$ to $\boldsymbol{S}$ in graphs $\mathcal{G}_{\boldsymbol{V}\backslash\boldsymbol{S}}$ in which directed edges into $\boldsymbol{S}$ are removed, that is $(\boldsymbol{S} \perp\!\!\!\perp \boldsymbol{R} \mid \boldsymbol{U})_{\mathcal{G}_{\boldsymbol{V}\backslash\boldsymbol{S}}}$. This holds for any causal diagram $\mathcal{G}$ compatible with an arbitrary $\mathcal{P}$. With these facts, we consider the following derivation to show that the proposed upperbound in smaller or equal to the natural upper bound,

$$\min_{\boldsymbol{z}} \left\{ \frac{Q[\boldsymbol{W}]}{\sum_{\boldsymbol{s},\boldsymbol{b}} Q[\boldsymbol{W}]} - \sum_{\boldsymbol{s}_k} \frac{Q[\boldsymbol{W}]}{\sum_{\boldsymbol{s},\boldsymbol{b}} Q[\boldsymbol{W}]} \right\} + Q[\boldsymbol{S}\backslash\boldsymbol{S}_k]$$

$$\leqslant \frac{Q[\boldsymbol{W}]}{\sum_{\boldsymbol{s},\boldsymbol{b}} Q[\boldsymbol{W}]} - \sum_{\boldsymbol{s}_k} \frac{Q[\boldsymbol{W}]}{\sum_{\boldsymbol{s},\boldsymbol{b}} Q[\boldsymbol{W}]} + Q[\boldsymbol{S}\backslash\boldsymbol{S}_k]$$

$$\overset{(1)}{\leqslant} Q[\boldsymbol{W}] - \sum_{\boldsymbol{s}_k} Q[\boldsymbol{W}] + Q[\boldsymbol{S}\backslash\boldsymbol{S}_k]$$

$$= P(\boldsymbol{w}) - P(\boldsymbol{s}\backslash\boldsymbol{s}_k, \boldsymbol{r} \mid do(\boldsymbol{v}\backslash\boldsymbol{w})) + P(\boldsymbol{s}\backslash\boldsymbol{s}_k \mid do(\boldsymbol{v}\backslash\boldsymbol{s}, \boldsymbol{s}_k))$$

$$= P(\boldsymbol{w}) + \sum_{\boldsymbol{u}} P(\boldsymbol{s}\backslash\boldsymbol{s}_k \mid \boldsymbol{u}, \boldsymbol{r}, do(\boldsymbol{v}\backslash\boldsymbol{w}))\{P(\boldsymbol{u}) - P(\boldsymbol{u},\boldsymbol{r})\}$$

$$\overset{(2)}{\leqslant} P(\boldsymbol{w}) + \sum_{\boldsymbol{u}} \{P(\boldsymbol{u}) - P(\boldsymbol{u},\boldsymbol{r})\}$$

$$= \sum_{\boldsymbol{v}\backslash\boldsymbol{w}} \{P(\boldsymbol{w}, \boldsymbol{v}\backslash\boldsymbol{w}) - P(\boldsymbol{r}, \boldsymbol{v}\backslash\boldsymbol{w})\} + 1$$

$$\overset{(3)}{\leqslant} P(\boldsymbol{w}, \boldsymbol{v}\backslash\boldsymbol{w}) - P(\boldsymbol{r}, \boldsymbol{v}\backslash\boldsymbol{w}) + 1$$

$$\overset{(4)}{=} P(\boldsymbol{x}, \boldsymbol{y}) - P(\boldsymbol{x}) + 1.$$

The last term being the expression of the natural lower bound. In the above, (1) holds since $Q[\boldsymbol{W}] - \sum_{\boldsymbol{s}_k} Q[\boldsymbol{W}] < 0$ and therefore multiplying by a number less than 1, namely $\sum_{\boldsymbol{s},\boldsymbol{b}} Q[\boldsymbol{W}]$, results in a larger expression; (2) holds by a similar observation since $P(\boldsymbol{u}) - P(\boldsymbol{u},\boldsymbol{r}) > 0$ and $P(\boldsymbol{s}\backslash\boldsymbol{s}_k \mid \boldsymbol{u}, \boldsymbol{r}, do(\boldsymbol{v}\backslash\boldsymbol{w})) < 1$; (3) holds since $P(\boldsymbol{w}, \boldsymbol{v}\backslash\boldsymbol{w}) - P(\boldsymbol{r}, \boldsymbol{v}\backslash\boldsymbol{w}) < 0$; (4) holds by definition since $\boldsymbol{W} \cup \boldsymbol{V}\backslash\boldsymbol{V} = \boldsymbol{X} \cup \boldsymbol{Y}$ and $\boldsymbol{R} \cup \boldsymbol{V}\backslash\boldsymbol{W} = \boldsymbol{X}$. □

**Prop. 6 restated**. *Partial IDP (Alg. 1) terminates and is sound.*

*Proof.* The proof follows from the termination guarantee of IDP, and the soundness of IDP (Jaber et al., 2019a) and Props. 3 and 4.

For the run time, let $n$ be the number of variables. Operations in the PID function of Alg. 1, such as computing *pc*-components or finding the set of possible ancestors or descendants (as done in Props. 3 and 4), could be done in $\mathcal{O}(n^2)$ time, e.g. with a Breadth-First Search algorithm. Line 10 in PID decomposes the input set $\boldsymbol{D}$ into at most $n$ subsets, each requiring a new call to PID. In turn, line 7 in PID, if triggered, will reduce the set $\boldsymbol{V}$ of size $n$ by at least one variable at the time resulting in at most $n$ additional separate calls to PID. Since line 7 might be triggered repeatedly for each decomposed *C*-factor in line 10, overall, Alg. 1 requires $\mathcal{O}(n^2)$ calls to PID and consequently $\mathcal{O}(n^4)$ time to return the bounds. □

**Prop. 7 restated.** *Given a PAG $\mathcal{P}$, let $\mathcal{L}$ be the set of ME LEGs. Then, $\mathbb{M}(\mathcal{P}) = \mathbb{M}(\mathcal{L})$.*

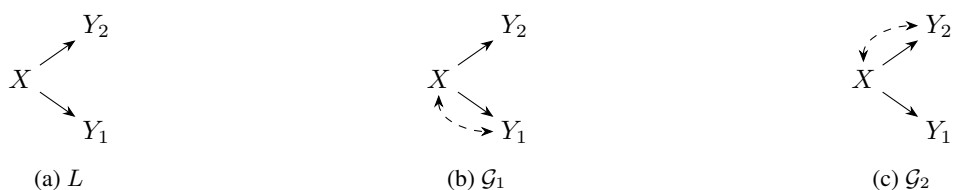

Figure 7: First example of MBD causal diagrams used in the proof of Prop. 9.

*Proof.* Each causal graph is a coarse representation of an underlying SCM, in which the presence of an edge $X \to Y$ implies the potential presence of $X$ as an argument in the function $f_Y$. In turn the absence of an edge implies a constraint in the system, e.g. no edge $X \to Y$ implies that $X$ is not an argument of $f_Y$. In general therefore the addition of an edge results in a family of SCMs that is strictly more general.

Each MAG can be converted to an LEG by replacing some of the bi-directed arrows by directed arrow. Bi-directed arrows exclude an ancestral relationship whereas directed arrows do not exclude unobserved confounding, a directed edge $X \to Y$ thus defines a strictly more general $f_Y$ in contrast with $X \leftarrow\!\text{-}\!\text{-}\!\text{-}\!\rightarrow Y$. Since this is the only difference between a MAG and its LEG, it follows that the set of SCMs compatible with a given MAG is strictly contained in the set of SCMs contained with its LEG. This reasoning applied to every MAG in the equivalence class implies that $\mathbb{M}(\mathcal{P}) = \mathbb{M}(\mathcal{L})$. □

**Prop. 8 restated.** *Given a PAG $\mathcal{P}$, let $\mathcal{D}$ be the set of ME MBD diagrams. Then, $\mathbb{M}(\mathcal{P}) = \mathbb{M}(\mathcal{D})$.*

*Proof.* For a given LEG $L \in \mathcal{L}$, let $\mathbb{G}_L$ be a set of maximally bi-directed causal graphs compatible with $L$. Then, We proceed by showing that $\mathcal{M}(\mathbb{G}_L) = \mathcal{M}(L)$.

Let $L$ be a given LEG, and write $\mathcal{G}_L$ for the causal graph with the same structure as $L$. Adding bi-directed edges, without breaking conditional independencies, leads to a graph $\mathcal{G}'_L$ with the same ancestral and conditional independencies as $L$. The difference between the constructed causal graph $\mathcal{G}'_L$ and $\mathcal{G}_L$ is that for at least one variable $X \in \boldsymbol{V}$, $f_{X,\mathcal{G}_L}(\boldsymbol{pa}_X, \boldsymbol{u}_X)$ while $f_{X,\mathcal{G}'_L}(\boldsymbol{pa}_X, \boldsymbol{u}_X, u_{X,Z})$, where $Z \in Pa_X$. The class of functions $f_X$ defined by $\mathcal{G}_L$ is included in that defined by $\mathcal{G}'_L$, and therefore $\mathbb{M}(\mathcal{G}_L) \subset \mathbb{M}(\mathcal{G}'_L)$.

In general, multiple graphs $\mathcal{G}'_L$ in which we repeatedly add bi-directed edges until no invisible edges exist can be constructed from a single LEG $L$. For example, the LEG $L := \{Y_1 \leftarrow X \to Y_2\}$ has two graphs can be constructed by adding bi-directed edges without removing statistical independencies: $\mathcal{G}_1 := \{Y_1 \leftarrow X \to Y_2, Y_1 \leftarrow\!\text{-}\!\text{-}\!\text{-}\!\rightarrow X\}$ and $\mathcal{G}_2 := \{Y_1 \leftarrow X \to Y_2, X \leftarrow\!\text{-}\!\text{-}\!\text{-}\!\rightarrow Y_2\}$. These graphs can be recovered exactly by Alg. 4. Let $\mathbb{G}_L$ be the set of all maximally bi-directed graphs compatible with $L$. Then, since any other causal graph compatible with $L$ defines a family of SCMs which is subsumed in that of a maximally directed causal graph, $\mathbb{M}(\mathbb{G}_L) = \mathbb{M}(L)$. □

**Prop. 9 restated.** *Given a PAG $\mathcal{P}$, let $\mathcal{G}$ and $\mathcal{H}$ be two MBD diagrams constructed from a ME LEG. In general, $\mathbb{M}(\mathcal{G}) \nsubseteq \mathbb{M}(\mathcal{H})$ and $\mathbb{M}(\mathcal{H}) \nsubseteq \mathbb{M}(\mathcal{G})$.*

*Proof.* We give explicit counterexamples to demonstrate this fact.

In general, the set of maximally bi-directed causal diagrams (MBD) compatible with a given LEG to consider for causal effect computation cannot be reduced without loss of generality. For instance, this could be shown for the computation of $P(y_1, y_2 \mid do(x))$ given the LEG $L$ in Fig. 7a. Here two MBD causal diagrams could be constructed: $\mathcal{G}_1$ and $\mathcal{G}_2$ in Fig. 7b and Fig. 7c respectively. Given that $P(y_1, y_2 \mid do(x)) = P(y_1 \mid do(x))P(y_2 \mid do(x))$ and that in $\mathcal{G}_1$: $P(y_1 \mid do(x)) \in [P(y_1, x), P(y_1, x) + 1 - P(x)], P(y_2 \mid do(x) = P(y_2 \mid x)$. (In this particular diagram, bounds could be derived analytically and are known to be provably tight (Pearl, 2009, Section 8.2). To further demonstrate this we provide below two SCMs compatible with $\mathcal{G}_1$ that evaluate to the upper and lower bounds respectively.). In contrast, in $\mathcal{G}_2$: $P(y_1 \mid do(x) = P(y_1 \mid x), P(y_2 \mid do(x)) \in [P(y_2, x), P(y_2, x) + 1 - P(x)]$. The causal effect differs across $\mathcal{G}_1$ and $\mathcal{G}_2$. Neither of the bounds computed from $\mathcal{G}_1$ or $\mathcal{G}_2$ include the other and therefore both have to be considered for correctly bounding causal effects from $L$.

Below we give two SCMs compatible with $\mathcal{G}_1$ whose causal effect $P(y_1 \mid do(x))$ evaluate to the lower and upper bounds

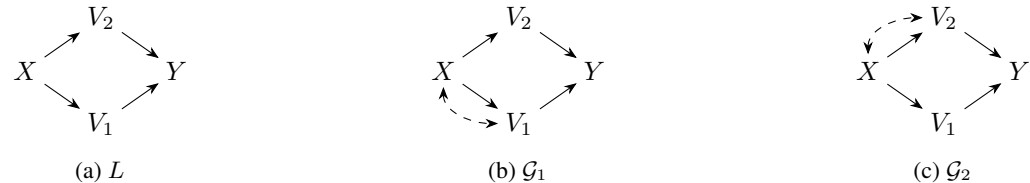

Figure 8: Second example of MBD causal diagrams used in the proof of Prop. 9.

obtained through posterior sampling illustrating the tightness of the returned bounds. Let $\mathcal{M}_1, \mathcal{M}_2 \in \mathbb{M}(\mathcal{G}_1)$ be defined by,

$$\mathcal{M}_1 := \begin{cases} x := f_X(u) \\ y_1 := \begin{cases} f_{Y_1}(x, u) & \text{if } x = f_X(u), \\ 0 & \text{otherwise.} \end{cases} \end{cases}$$

and,

$$\mathcal{M}_2 := \begin{cases} x := f_X(u) \\ y_1 := \begin{cases} f_{Y_1}(x, u) & \text{if } x = f_X(u), \\ 1 & \text{otherwise.} \end{cases} \end{cases}$$

Assume further that $P_{\mathcal{M}_1}(u) = P_{\mathcal{M}_2}(u)$. Then, both SCMs agree on observational distributions $P_{\mathcal{M}_1}(x, y_1) = P_{\mathcal{M}_2}(x, y_1) = P(x, y_1)$. However the following derivations show that the interventional distribution $P(y_1 = 1 \mid do(x = 1))$ differs across models: for $\mathcal{M}_1$ equal to the analytical lower bound, and for $\mathcal{M}_2$ equal to the analytical upper bound demonstrating that (in this case) the bound is tight. In particular,

$P_{\mathcal{M}_1}(y_1 = 1 \mid do(x = 1))$
$\quad = P_{\mathcal{M}_1}(y_1 = 1 \mid x = 1, u : x = f_X(u))P(u : x = f_X(u)) + P_{\mathcal{M}_1}(y_1 = 1 \mid x = 1, u : x \neq f_X(u))P(u : x \neq f_X(u))$
$\quad = P(y_1 = 1 \mid x = 1)P(x = 1)$
$\quad = P(y_1 = 1, x = 1),$
$P_{\mathcal{M}_2}(y_1 = 1 \mid do(x = 1))$
$\quad = P_{\mathcal{M}_2}(y_1 = 1 \mid x = 1, u : x = f_X(u))P(u : x = f_X(u)) + P_{\mathcal{M}_2}(y_1 = 1 \mid x = 1, u : x \neq f_X(u))P(u : x \neq f_X(u))$
$\quad = P(y_1 = 1 \mid x = 1)P(x = 1) + P_{\mathcal{M}_2}(y_1 = 1 \mid x = 1, u : x \neq f_X(u))P(u : x \neq f_X(u))$
$\quad = P(y_1 = 1, x = 1) + 1 - P(x = 1).$

This holds also more generally for single output causal effect of the form $P(y \mid do(x))$. For instance, bounding $P(y \mid do(x))$ given the LEG $L$ in Fig. 8a requires two MBD causal diagrams without loss of generality, given in Fig. 8b and Fig. 8c respectively. It could be shown by writing $P(y \mid do(x)) = \sum_{v_1, v_2} P(y \mid x, v_1, v_2)P(v_1, v_2 \mid do(x))$ that the upper bounds computed from $\mathcal{G}_1$ and $\mathcal{G}_2$ will disagree as upper bounds for $P(v_1, v_2 \mid do(x))$ will differ (as shown in the example above). $\qquad\square$

## C ADDITIONAL EXAMPLES

We provide in this section additional examples to illustrate the proposed bounding procedure.

**Example 4.** In this example we illustrate the quantities mentioned in Props. 3 and 4 by applying these propositions to bound $Q[Y]$ from $Q[W, X, Z, Y] = P(w, x, z, y)$ given the PAG $\mathcal{P}$ in Fig. 2. Using the notation in Props. 3 and 4, we can write $\boldsymbol{W} = \text{PossAn}(Y)_{\mathcal{P}_{W,X,Z,Y}} = \{W, X, Z, Y\}$ and $\boldsymbol{R} = \{W, X, Z, Y\} \backslash \{Y\} = \{W, X, Z\}$ which can be partitioned into $\boldsymbol{A} = \{W\}, \boldsymbol{B} = \{X, Z\}$ since $\{W\}$ is not a possible descendant of any possible spouse of $Y$ in $\mathcal{P}$. $\text{PossPa}(\boldsymbol{W}) \backslash \text{PossPa}(Y) = \{W\}$. The lower bound then follows by replacing these variables in the statement of the lower bound to get,

$$Q[Y] \geqslant \max_w \frac{Q[W, X, Z, Y]}{\sum_{z,x,y} Q[W, X, Z, Y]}. \tag{27}$$

Equivalently $P_{x,w,z}(y) \geqslant \max_w P(z, y, x \mid w)$. Moreover,

$$Q[Y] \leqslant 1 + \min_w \left\{ \frac{Q[W, X, Z, Y]}{\sum_{z,y,x} Q[W, X, Z, Y]} - \sum_y \frac{Q[W, X, Z, Y]}{\sum_{z,y,x} Q[W, X, Z, Y]} \right\},$$

or equivalently,

$$P_{x,w,z}(y) \leqslant \min_w \{P(z, y, x \mid w) - P(z, x \mid w)\} + 1. \tag{28}$$

We could show, moreover, that these bounds are tighter than the natural bounds as $P(y, z, x, w) = P(y, z, x \mid w)P(w) \leqslant P(y, z, x \mid w) \leqslant \max_w P(y, z, x \mid w) \leqslant P_{z,x,w}(y)$ and as,

$$P(y, z, x, w) - P(z, x, w) + 1 \geqslant 1 + \sum_w P(y, z, x, w) - P(z, x, w)$$

$$= \sum_w P(w) \Big\{ P(y, z, x \mid w) - P(z, x \mid w) + 1 \Big\}$$

$$\geqslant \sum_w P(w) \min_w \Big\{ P(y, z, x \mid w) - P(z, x \mid w) + 1 \Big\}$$

$$= \min_w \Big\{ P(y, z, x \mid w) - P(z, x \mid w) \Big\} + 1. \tag{29}$$

∎

**Example 5.** Consider the query $P_{x_1, x_2}(y)$ given the PAG $\mathcal{P}$ in Fig. 9a. We will proceed by decomposing the query into smaller components and either uniquely identify or bound each term as appropriate with Alg. 1. Using the $c$-factor notion $Q[\cdot]$, we have that $P_{x_1, x_2}(y) = \sum_{a,b,c} Q[Y, A, B, C]$ by line 1 since the set $\{Y, A, B, C\}$ is ancestral in the sub-graph $\mathcal{P}_{\{Y,A,B,C,W\}}$ (see also Lem. 2 in Appendix B for further justification). Calling IDP on this component with $\boldsymbol{T} = \boldsymbol{V}$ and $\boldsymbol{C} = \{Y, A, B, C\}$, a first simplification can be done in line 6 as the if condition is triggered for $\boldsymbol{B} = \{X_1\}$ which updates $\boldsymbol{T}$ to $\boldsymbol{T} = \{Y, A, B, C, W, X_2\}$. Successive calls to IPD trigger line 9, where we find in a first instance that $Q[Y, A, B, C] = Q[A]Q[Y, B, C]$ as the region of $A$ in $\mathcal{P}_{\{A,Y,B,C\}}$ is $\mathcal{R}_A^{\{A,Y,B,C\}} = \{A\}$, the region of $\{Y, B, C\}$ in the same sub-graph is $\mathcal{R}_{\{Y,B,C\}}^{\{A,Y,B,C\}} = \{Y, B, C\}$, and have an empty intersection. In a second instance, we find that $Q[Y, B, C] = Q[Y]Q[B, C]$ where $Q[B, C] = \sum_{x_1, w} Q[W, X_1, B, C]$. $Q[W, X_1, B, C]$ is identifiable from $\mathcal{P}$ in line 7 of Alg. 1 which returns $P(c \mid a, x_1, b, w)P(x_1, b, w)$.

Finally, $\boldsymbol{C}$ reduces to $\boldsymbol{C} = \{Y\}$ after these simplifications. With an additional call to IDP we find that $\boldsymbol{T}$ could be reduced to $\boldsymbol{T} = \{X_2, Y\}$ in line 6, and further that $Q[X_2, Y] = P(y \mid x_2, b, c)P(x_2)$. Now, no more simplifications could be done as calls to lines 6 and 9 in Alg. 1 fail, due to the potential presence of an unobserved confounder between $X_2$ and $Y$. In line 16, we therefore proceed to bound $Q[Y]$ from $Q[X_2, Y]$ using Props. 3 and 4. We find that,

$$Q[Y, X_2] \leqslant Q[Y] \leqslant Q[Y, X_2] - \sum_y Q[Y, X_2] + 1 \tag{30}$$

Finally, putting each term in the expression $P_{x_1, x_2}(y) = \sum_{a,b,c} Q[A]Q[Y]Q[B, C]$ together we have the lower bound is:

$$P_{x_1, x_2}(y) \geqslant \sum_{a,b,c} P(y \mid x_2, b, c)P(x_2)P(a \mid x_1) \sum_{w,x_1} P(w, b, x_1)P(c \mid a, w, b, x_1), \tag{31}$$

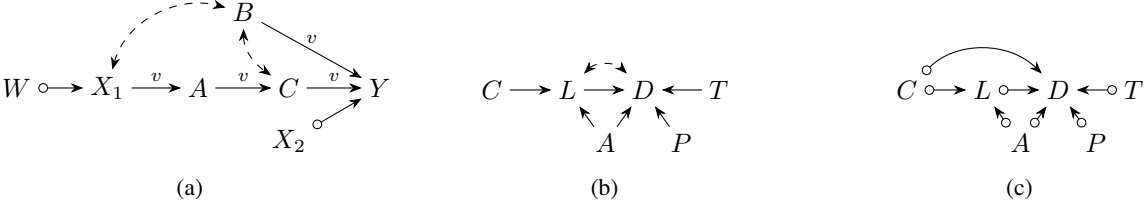

(a)           (b)           (c)

Figure 9: (a) PAG for Example 5, (b) causal diagram and (c) PAG for Example 6.

and the upper bound is:

$$P_{x_1,x_2}(y) \leqslant \sum_{a,b,c} \left(P(y \mid x_2, b, c)P(x_2) + 1 - P(x_2)\right) P(a \mid x_1) \sum_{w,x_1} P(w, b, x_1) P(c \mid a, w, b, x_1). \tag{32}$$

∎

**Example 6** (Applications in public health). As an additional example, we consider an adaptation of the Lung Cancer diagram from (Lauritzen and Spiegelhalter, 1988), shown in Fig. 9b. We are interested in determining the effect of a chemotherapy treatment ($C$) for lung cancer ($L$) on dyspnoea ($D$), *i.e.* breathing difficulty, in the context of other factors such as smoking (unobserved and represented with a bi-directed edge), pollution ($P$), age ($A$), and Tuberculosis ($T$). For this experiment, we generate $1,000$ samples from a compatible SCM.

The PAG inferred from data is given in Fig. 9c. Given the uncertainty in the underlying causal relations between variables, we could only partially identify the causal effect of interest in this example with Alg. 1. The first step is to leverage the possible ancestors of $D$ to write: $P(d \mid do(c)) = \sum_{l,a,p,t} Q[D, L, A, P, T]$. A call to PID with this term reveals that no further simplification can be made, and a as a consequence we proceed to bound $Q[D, L, A, P, T]$ from $Q[\boldsymbol{V}$ with Props. 3 and 4. We obtain that,

$$P(d \mid do(c)) \geqslant \sum_{l,a,p,t} Q[D, L, A, P, T, C] = P(d, c) \tag{33}$$

and that,

$$P(d \mid do(c)) \leqslant \sum_{l,a,p,t} Q[D, L, A, P, T, C] + 1 - \sum_{l,a,p,t} Q[C, L, A, P, T] = P(d, c) + 1 - P(c) \tag{34}$$

These expressions imply that knowledge of the PAG and observational data license the following approximate bounds,

$$P(d = 1 \mid do(c = 1)) \in [0.5087, 0.9353]. \tag{35}$$

As a contrast, if we were to evaluate the causal effect assuming the causal diagram in Fig. 9b, the causal effect can be approximated to be $P(d = 1 \mid do(c = 1)) = P(d = 1 \mid c = 1) = 0.7775$. ∎

The Lung Cancer data is generated from the following SCM compatible with the consensus causal diagram given in Fig. 9b. In the following, $\mathbb{1}\{\cdot\}$ is the indicator function that equals 1 if the statement in $\{\cdot\}$ is true, and equal to 0 otherwise. $P(u_C), P(u_L), P(u_A), P(u_D), P(u_P), P(u_T)$ are given by independent Gaussian distributions with mean 0 and variance 1, and each observation $(c, l, a, d, p, t)$ generated from exogenous variables using the structural assignments: $c \leftarrow \mathbb{1}\{u_C < 0\}, l \leftarrow \mathbb{1}\{c - u_L > 0\}, a \leftarrow \mathbb{1}\{u_A > 0\}, d \leftarrow \mathbb{1}\{l + a - 2u_d + 0.2p - 0.3T > -0.5\}, p \leftarrow \mathbb{1}\{u_P > 0\}, t \leftarrow \mathbb{1}\{u_T > 0\}$. The ground truth value of the causal effect is given by setting $c \leftarrow 1$ and evaluating $P(d = 1)$ under this updated model.

---
**Algorithm 3** Depth-first search
---
 1: **Input:** LEG $L$
 2: **Output:** Set of Markov equivalent LEGs

 3: Initialize $S$ as an empty list and append $L$
 4: **return** $\texttt{Search}(L, S)$

 5: **function** $\texttt{Search}(L, S)$
 6:    **for** each $L'$ obtained by performing a legitimate edge reversal **do**
 7:       If $L' \notin S$, append $L'$ to $S$
 8:       Run $\texttt{Search}(L', S)$
 9:    **end for**
10: **return** $S$

---

# D   FURTHER DISCUSSION ON ENUMERATION STRATEGIES

One approach for enumerating Markov equivalent LEGs can be derived from the transformational characterization in Prop. 10 by reversal of directed edges. Similarly to how one could enumerate all Markov equivalent DAGs with covered edge reversals as done by Wienöbst et al. (2023), the following depth-first-search program could be used for LEGs.

1. Let $S$ be an empty list

2. Append $L$ to $S$

3. Run **search**$(L, S)$

Starting from an LEG $L$, all neighbors of $L$ (graphs with a single reversed edge) are explored and this is continued recursively. Eventually all ME LEGs are reached by Prop. 10 above and the algorithm terminates. In order to not visit any LEG twice, it is necessary to store a set of all visited LEGs. An algorithm implementing this procedure is given in Alg. 3.

**Proposition 11.** *Alg. 3 enumerates a Markov equivalence class of LEGs and can be implemented with worst-case delay* $\mathcal{O}(m^3)$, *where $m$ is the number of edges in any LEG of the equivalence class.*

*Proof.* The proof of (Wienöbst et al., 2023, Theorem 9) applies to the case of LEGs as the space of ME LEGs can be traversed by a sequence of edge traversals (starting from any LEG) as shown by Zhang and Spirtes (2012b).

$\square$

Alg. 4 retrieves the set of MBD causal diagrams compatible with a given LEG. It proceeds by adding dashed bi-directed edges $X \dashleftarrow\dashrightarrow Y$ for every invisible edge between $X$ and $Y$. When bi-directed edges for two adjacent invisible edges cannot be added without violating a conditional independence, two diagrams are constructed and the process continues along two separate branches of the tree. For example, $\{X \to Z \to Y\}$ creates $\{X \to Z \to Y, X \dashleftarrow\dashrightarrow Z\}$ and $\{X \to Z \to Y, Z \dashleftarrow\dashrightarrow Y\}$ separately as $\{X \to Z \to Y, X \dashleftarrow\dashrightarrow Z, Z \dashleftarrow\dashrightarrow Y\}$ violates the conditional independence between $X$ and $Y$.

**Proposition 12.** *Alg. 4 enumerates all MBD causal diagrams compatible with a given LEG.*

*Proof.* In line (11), Alg. 4 appends and returns causal diagrams that can be constructed from a given LEG by adding the maximum number of bi-directed edges, *i.e.* diagrams without invisible edges and that therefore exclude any further unobserved confounding. Lines (9) and (12) ensure that all possible diagrams to which a bi-directed edge could be added are considered and ensures that all intermediate diagrams with invisible edges are maintained for analysis. At termination therefore Alg. 4 will have enumerated all MBD causal diagrams compatible with a given LEG. $\square$

---

**Algorithm 4** Derivation of maximally bi-directed causal graphs

---

1: **Input:** LEG $L$
2: **Output:** The set of maximally bi-directed graphs.
3: For all invisible edges of the form $X \rightarrow Y$ such that $X$ does not have any parents or spouses not adjacent to $Y$, add a bi-directed edge $X \leftarrow\text{-}\text{-}\text{-}\text{-}\rightarrow Y$ to $L$, creating a causal diagram $G$

4: Initialize an empty list $S$ and append $\mathcal{G}$ to it
5: Initialize an empty list $S'$

6: **while** $S$ is not empty **do**
7:     **for** each $\mathcal{G}$ in $S$ **do**
8:         Let $k$ be the number of invisible edges in $\mathcal{G}$
9:         Create graphs $\mathcal{G}_1, \ldots, \mathcal{G}_k$ by adding a bi-directed edge for each invisible edge in $L$
10:         Mark visible edges in $\mathcal{G}_1, \ldots, \mathcal{G}_k$
11:         Append $S'$ with the subset of $\{\mathcal{G}_1, \ldots, \mathcal{G}_k\}$ in which invisible edges do not exist
12:         Append $S$ with the subset of $\{\mathcal{G}_1, \ldots, \mathcal{G}_k\}$ in which invisible edges do exist and remove $\mathcal{G}$
13:     **end for**
14: **end while**
15: **return** $S'$

---

