# OpenReview forum: "Towards Bounding Causal Effects under Markov Equivalence"
_auai.org/UAI/2024/Conference — UAI 2024 oral_

### Official Review · Reviewer_xRqA · 2024-03-05

**Q2-1 Originality-Novelty:** 3
**Q2-2 Correctness-Technical Quality:** 2
**Q2-5 Clarity Of Writing:** 2

**Q1 Summary And Contributions:**

This paper derives bounds on causal effects within a Markov equivalence class of causal graphs.

**Q2-3 Extent To Which Claims Are Supported By Evidence:**

3: Good: the main claims are supported by convincing evidence (in the form of adequate experimental evaluation, proofs, (pseudo-)code, references, assumptions).

**Q2-4 Reproducibility:**

3: Good: key resources (e.g. proofs, code, data) are available and key details (e.g. proofs, experimental setup) are sufficiently well-described for competent researchers to confidently reproduce the main results.

**Q3 Main Strengths:**

The challenge of estimating causal effects without concrete knowledge of the underlying causal structure is highly relevant. The idea of deriving bounds for causal effects within a Markov equivalence seems promising and straightforward.

**Q4 Main Weakness:**

Unfortunately, I found it quite hard to follow the presentation of the paper. While I understand the wish to shorten notation, some math displays are really hard to follow/not mathematically precise/correct (e.g. eq (3) sum over some lower case c, the expression inside the sum does not contain lower case c, but another sum with index lower case c).

**Q5 Detailed Comments To The Authors:**

In the abstract, you talk about a 'data-driven setting'. As far as I understood, the bounds are based on population-level distributions and do not incorporate statistical uncertainty. Is there a way to give probabilistic guarantees for bounds based on finite data?

**Q9 Complying With Reviewing Instructions:**

Yes

---

> ### Author Rebuttal · Authors · 2024-04-04
>
> Thank you for your time, we appreciate the feedback. In the following we attempt to address your concerns on the mathematical notation and probabilistic guarantees with finite samples.
>
> ***"While I understand the wish to shorten notation, some math displays are really hard to follow/not mathematically precise/correct (e.g. eq (3) sum over some lower case c, the expression inside the sum does not contain lower case c, but another sum with index lower case c).”***
>
> We sympathize with this comment. We inherit the $C$-factor notation $Q[\cdot]$, and the notation for describing operations over $C$-factors from the literature, see e.g. Sec. 4.2 in (Tian and Pearl, 2003) and specifically Proposition 4 in (Jaber et al, 2019b). We apologize if it was found to be confusing. To clarify, in Eq. (3), $Q[\boldsymbol C \cup \boldsymbol Z]:= P\_{\boldsymbol v \backslash (\boldsymbol c \cup \boldsymbol z)}(\boldsymbol c, \boldsymbol z)$.  The inner sum then sums this interventional probability over values $\boldsymbol c$ of $\boldsymbol C$ while the outer sum then sums the resulting ratio over values $\boldsymbol c \backslash \boldsymbol y$ of $\boldsymbol C \backslash \boldsymbol Y$. We will recall this notation shorthand to be more explicit, thank you.
>
> ***”In the abstract, you talk about a 'data-driven setting'. As far as I understood, the bounds are based on population-level distributions and do not incorporate statistical uncertainty. Is there a way to give probabilistic guarantees for bounds based on finite data?”***
>
> The setting is said to be “data-driven” because it does not require the correct specification of a causal graph, and instead leverages the independencies that could be found in data (assuming those relate closely to $d$-separations in the underlying causal graph). Considering the estimation of bounds from finite samples is indeed an important consideration in practice. At this moment, we could provide some initial pointers for future research. One promising avenue, given that the correct PAG is given, is to extend the estimation procedure of (Jung et al, 2020). The authors showed that each $C$-factor $Q[\cdot]$ may be written as a weighted distribution leading to a version of the ID algorithm (Tian et al, 2002) that identifies a causal effect as a weighted version of the observational distribution. As the proposed Partial IDP algorithm (Alg. 1) similarly involves combinations of $C$-factors, we conjecture that one might express (a version of) the proposed bounds as a weighted distribution that is more amenable to estimation from finite samples. With this procedure, one could then explore deriving confidence intervals around probabilities defined with estimated weights, use the non-parametric bootstrap, etc. We should note also that inferring the correct PAG from finite samples is difficult (see also Appendix A.3). Errors in the estimated PAG may propagate to the estimation of bounds and generally stronger assumptions, e.g. strong faithfulness, would be required for consistent estimation in practice.
>
> Jung, Yonghan, Jin Tian, and Elias Bareinboim. "Learning causal effects via weighted empirical risk minimization." Advances in neural information processing systems 33 (2020): 12697-12709.
>
> Jin Tian and Judea Pearl. “A general identification condition for causal effects.“ eScholarship, University of California, 2002.

---

### Official Review · Reviewer_WCTd · 2024-03-19

**Q2-1 Originality-Novelty:** 3
**Q2-2 Correctness-Technical Quality:** 3
**Q2-5 Clarity Of Writing:** 4

**Q1 Summary And Contributions:**

The paper presents results on bounding causal effects when the PAG is known. The paper presents two approaches: In the first part, the authors derive bounds for a causal effect by essentially looking into possible configurations of the region of the effect of interest. The authors give an algorithm that takes a PAG and a query as input, and returns the causal effect, if identifiable, or a bound otherwise. The authors show that these bounds are at least as tight as the natural bounds. The methods is shown to be sound.  In the second part,  the authors focus on the existing approach of bounding causal effects through enumerating possible MAGs/SCMs consistent with a PAG. To that effect, the authors show that some ME SCMs  do not to be included in the enumeration.

**Q2-3 Extent To Which Claims Are Supported By Evidence:**

3: Good: the main claims are supported by convincing evidence (in the form of adequate experimental evaluation, proofs, (pseudo-)code, references, assumptions).

**Q2-4 Reproducibility:**

4: Excellent: key resources (e.g. proofs, code, data) are available and key details (e.g. proof sketches, experimental setup) are comprehensively described for competent researchers to confidently and easily reproduce the main results.

**Q3 Main Strengths:**

-The paper presents a novel sound algorithm for bounding causal effects when the PAG is known.
-The authors give some examples where the bounds are quite informative.

**Q4 Main Weakness:**

-The complexity of deriving the bounds described in Propositions 3 and 4 is not discussed.
-The presentation of the enumeration section is not as clear.

**Q5 Detailed Comments To The Authors:**

I believe that this is a nice paper with a novel contribution that should be accepted.
I would like some discussion on the complexity of Algorithm 1. I also found that Section 5 was not as clear as the rest of the paper, perhaps an algorithm for bounding causal effects using enumeration would make the results clearer

**Q9 Complying With Reviewing Instructions:**

Yes

---

> ### Author Rebuttal · Authors · 2024-04-04
>
> Thank you for your thoughtful review and constructive feedback. In the following, we hope to provide an approximate description of the complexity of Alg. 1 by following similar derivations proposed for the ID algorithm (for point-identifying causal effects) and make a comment on Sec. 5.
>
> Let $n$ be the number of variables. Operations in the PID function of Alg. 1, such as computing $pc$-components or finding the set of possible ancestors or descendants (as done in Props. 3 and 4), could be done in $\mathcal O(n^2)$ time, e.g. with a Breadth-First Search algorithm. Line 10 in PID decomposes the input set $\boldsymbol D$ into at most $n$ subsets, each requiring a new call to PID. In turn, line 7 in PID, if triggered, will reduce the set $\boldsymbol V$ of size $n$ by at least one variable at the time resulting in at most $n$ additional separate calls to PID. Since line 7 might be triggered repeatedly for each decomposed $C$-factor in line 10, overall, Alg. 1 requires $\mathcal O(n^2)$ calls to PID and consequently $\mathcal O(n^4)$ time to return the bounds.
>
> As a note on Sec. 5, its purpose is to show that a potentially large number of causal graphs must be considered by methods that bound causal effects given a specific graph. We did not go into details as to exactly how “graph-specific” methods go about bounding causal effects. To better contextualize that section of the paper, and for completeness, we will go into more details in the related work section of the Appendix.

---

### Official Review · Reviewer_G4Wq · 2024-03-22

**Q2-1 Originality-Novelty:** 3
**Q2-2 Correctness-Technical Quality:** 3
**Q2-5 Clarity Of Writing:** 3

**Q1 Summary And Contributions:**

In this paper, the authors provide the bounds of causal effect by only using observational data in the presence of unobserved confounding, which can be represented by a PAG. Moreover, the authors demonstrate in theory that the proposed boundary is no worse than the natural boundary.

**Q2-3 Extent To Which Claims Are Supported By Evidence:**

3: Good: the main claims are supported by convincing evidence (in the form of adequate experimental evaluation, proofs, (pseudo-)code, references, assumptions).

**Q2-4 Reproducibility:**

3: Good: key resources (e.g. proofs, code, data) are available and key details (e.g. proofs, experimental setup) are sufficiently well-described for competent researchers to confidently reproduce the main results.

**Q3 Main Strengths:**

Under fewer assumptions, the authors give the existing methods as potentially better bounds

**Q4 Main Weakness:**

See Q5

**Q5 Detailed Comments To The Authors:**

0. In this article, w, r are fixed, satisfy those three conclusions, and the computed bounds are deterministic; is it possible that there are other set choices that also satisfy those conclusions and then get tighter bounds?

1.I have some confusion about the definition of “partial identification”, that is, Definition 2 and Definition 6. First, for Definition 2, is it possible that causal effects are not partially identifiable? If so, can you give an example? If not, then “partial identification” holds naturally for any causal effect, which makes this definition meaningless. The same question applies to Definition 6. Second, what's the difference between Definition 2 and Definition 6? Why do you give two different definitions?

2.	The definition of MAG uses “m-separation” instead of “d-separation”, and the former is a natural extension of the latter.

3.	In Example 2, the author said “The pc-component that contains Y is given by {W, X, Z, Y}”. Why are X and Y in the same pc-component?

4.	In Example 3, the author said “{RAF, MEK, ERK,PKA} is ancestral in the corresponding PAG.” Why? PKC is a possible ancestor of RAF, isn’t it?

**Q9 Complying With Reviewing Instructions:**

Yes

---

> ### Author Rebuttal · Authors · 2024-04-04
>
> Thank you for your thoughtful review, we appreciate the positive feedback. In the following, we hope to provide clarifications and additional explanations to address mentioned weaknesses and questions.
>
> ***0. "In this article, w, r are fixed, satisfy those three conclusions, and the computed bounds are deterministic; is it possible that there are other set choices that also satisfy those conclusions and then get tighter bounds?”***
>
> Thank you for your question. It is correct that $\boldsymbol W, \boldsymbol R$ are fixed by the definition of $\boldsymbol C$ and $\boldsymbol S$  in Props. 3 and 4, although in principle several choices of $\boldsymbol C$ may lead to correct bounds (as long as $Q[\boldsymbol C]$ is identifiable and $\boldsymbol S \subset \boldsymbol C)$. In particular, larger $\boldsymbol C$ will lead to looser bounds and smaller $\boldsymbol C$ will lead to tighter bounds. One version of your question is then whether a smaller $\boldsymbol C$ exists than that considered in Alg. 1 when executing step 16. This cannot be the case as the decompositions in steps 6-14 ensure that every $C$-factor is as small as possible. Further, with our current proof strategy we believe to have chosen $\boldsymbol W$ and $\boldsymbol B$ to lead to the tightest possible bounds (although we cannot exclude that tighter bounds could be derived with a different bounding strategy).
>
> ***1. ”I have some confusion about the definition of “partial identification”, that is, Definition 2 and Definition 6. First, for Definition 2, is it possible that causal effects are not partially identifiable? If so, can you give an example? If not, then “partial identification” holds naturally for any causal effect, which makes this definition meaningless. The same question applies to Definition 6. Second, what's the difference between Definition 2 and Definition 6? Why do you give two different definitions?”***
>
> Regarding Def. 2, a priori it is not immediate that the observational distribution would force causal effects to lie in a sub-interval of $[0,1]$, i.e. that causal effects are partially identifiable. In this context, defining the notion of partial identification can be useful to the reader (especially if mentioned before introducing the natural bounds) even if later it can be shown that causal effects are naturally identifiable. A second reason is to be able to more directly relate bounds with and without the PAG in Prop. 2. Defs. 2 and 6 are useful for that purpose as they explicitly define the notion of partial identifiability of causal effects with respect to structural causal models that can be restricted to be compatible with the PAG or not.
>
> ***2. ”The definition of MAG uses “m-separation” instead of “d-separation”, and the former is a natural extension of the latter.”***
>
> We will make this correction, thank you.
>
> ***3. "In Example 2, the author said “The pc-component that contains Y is given by {W, X, Z, Y}”. Why are X and Y in the same pc-component?"***
>
> In the PAG in Fig. 2, $X$ and $Y$ are *not* in the same $pc$-component (as there is no path between them made of colliders and in which none of the edges are visible). Thank you for pointing this out with your question. In any case, this sentence does not play a role in the example; it can be removed without changing any part of the derivation. To summarize the key steps of Example 2 for context, we seek to bound $Q[Y]$ from the joint distribution $P(\boldsymbol v)$ and the PAG: the first step involves identifying $Q[W,X,Z,Y]$ from $Q[W,X,Z,Y,S]=:P(\boldsymbol v)$ (line 7 of Alg. 1), then, second, noticing that $Q[W,X,Z,Y]$ cannot be reduced further (the two if statements are not triggered), we bound $Q[Y]$ from $Q[W,X,Z,Y]$ using Props. 3 and 4 in line 16.
>
> ***4. "In Example 3, the author said “{RAF, MEK, ERK,PKA} is ancestral in the corresponding PAG.” Why? PKC is a possible ancestor of RAF, isn’t it?"***
>
> By ``corresponding PAG’’, we mean the subgraph restricted to the variables $(\texttt{RAF, MEK, ERK, PKA, AKT})$, i.e. $\mathcal P_{(\texttt{RAF, MEK, ERK, PKA, AKT})}$. We mention this is in the example to point out that since $(\texttt{RAF, MEK, ERK, PKA})$ is ancestral in this subgraph, we have that $\sum_{\texttt{AKT}} Q[\texttt{RAF, MEK, ERK, PKA, AKT}] = Q[\texttt{RAF, MEK, ERK, PKA}]$ by Lemma 2 in the Appendix.

---

### Official Review · Reviewer_7Gng · 2024-03-23

**Q2-1 Originality-Novelty:** 4
**Q2-2 Correctness-Technical Quality:** 3
**Q2-5 Clarity Of Writing:** 4

**Q1 Summary And Contributions:**

To summarize the paper, here's some brief (if slightly misleading) background.  Scientists now know how to use *experimental* data to estimate causal effects.  If one is interested in what I'll call "unconditional" effects (i.e., where an intervention is performed but the conditioning set is empty), standard methods in the potential outcomes framework allow the calculation of causal effects from experimental data (or by use of so-called instrumental variables).  If one is interested in conditional effects, one needs to specify a more complex causal model - like an SCM - and perform some calculations in the do-calculus or something similar.  Importantly, both these approaches more-or-less assume that one has access to experimental data that determines the directions of all relevant edges in one's causal model.

Although there is an extensive literature on causal discovery from observational data, less work has been done on *estimating causal effects* using only observational data.  That is the task of this paper.

**Q2-3 Extent To Which Claims Are Supported By Evidence:**

4: Excellent: all claims are supported by very convincing evidence (in the form of comprehensive experimental evaluation, rigorous mathematical proofs, detailed (pseudo-)code, precise references, well-motivated and realistic assumptions) and the authors deliver what they promise.

**Q2-4 Reproducibility:**

4: Excellent: key resources (e.g. proofs, code, data) are available and key details (e.g. proof sketches, experimental setup) are comprehensively described for competent researchers to confidently and easily reproduce the main results.

**Q3 Main Strengths:**

Overall, this is an excellent paper, in my opinion.  It contains a combination of theoretical, computational, and empirical results.  Theoretically, the paper shows there are non-trivial causal effects even when the underlying graphical model is a PAG.  Computationally, the paper shows how such bounds can be calculated. And finally, the paper gives non-trivial empirical applications of those results.

The writing in the paper is also fairly clear, and the results are well-organized an dmotivated.

**Q4 Main Weakness:**

As I see things, the authors acknowledge the main weakness of their work only briefly on page 7.  There, they write, "Note, however, that these bounds do not account for statistical uncertainty: a more careful analysis would be required to make actionable causal claims."

To me, that acknowledgement comes a bit too late in the paper.   The author's acknowledge their results rely on faithfulness.  But In the absence of stronger faithfulness assumptions, no non-trivial bounds on causal effects are achievable from purely observational data once one accounts for statistical uncertainty.  See

Robins, J., Scheines, R., Spirtes, P., and Wasserman, L. (1999) “Uniform Consistency in Causal Inference,” Biometrika 90:491-515.

Kelly, K. T., & Mayo-Wilson, C. (2010). Causal conclusions that flip repeatedly and their justification. Proceedings of the Twenty-Sixth Conference on Uncertainty in Artificial Intelligence, (pp. 277–286).

**Q5 Detailed Comments To The Authors:**

Again, this is a great paper, and so I have few suggestions except the one mentioned on my response to "main weaknesses."

**Q9 Complying With Reviewing Instructions:**

Yes

---

> ### Author Rebuttal · Authors · 2024-04-04
>
> We really appreciate the positive assessment of our work, thank you. Below, we make a brief comment on your observation that stronger assumptions would typically be required for inference from finite samples.
>
> Firstly, thank you for sharing these references on faithfulness and causal discovery with finite samples. In them, the authors highlight that causal discovery cannot be reliable on a given sample size under faithfulness: in our case, therefore, we could expect that errors in the estimated equivalence class (PAG) may lead to invalid bounds on causal effects computed from that estimated PAG. This estimation challenge is highly non-trivial, as mentioned by the reviewer, although in general it can be separated from the identification challenge, i.e. inferring an expression to bound causal effects, which is the focus of our paper. See also the discussion in Appendix A.3 where this limitation is described in more details. We will make this point earlier in the paper and acknowledge more explicitly that for the validity of bounds from PAGs estimated from finite samples one should consider stronger faithfulness assumptions for which the inference of causal structure can be uniformly consistent (Zhang and Spirtes, 2003).
>
> Zhang, Jiji, and Peter L. Spirtes. "Strong faithfulness and uniform consistency in causal inference." arXiv preprint arXiv:1212.2506 (2012).

---

### Meta-Review · Area_Chair_fe4B · 2024-04-16

This paper considers total causal effect identification given a PAG. In particular, the authors discuss bounds on the causal effect when this effect is not pointwise identifiable using Jaber et al.'s 2020 ID algorithm.
The reviewers are in agreement that this paper deserves acceptance to the UAI conference track, with an average score of 7.75. This is an exciting and underdeveloped research direction that deserves more attention.

- The reviews of the paper are positive overall, with some minor comments that the authors have addressed in their rebuttal.
I would request that the authors address the reviewers' comments in the manuscript as they did in their author response.

I also have a few comments that I would appreciate the authors reflecting on.

- Section 5 discusses the difficulty of enumerating MAGs represented by a PAG as an alternative strategy for deriving tighter bounds on identification. I think this section is missing a few key references and some discussion.
Firstly, in the absence of latent variables, equivalent versions of this approach called IDA-type algorithms (see Maathuis et al. 2009)  have been developed to obtain a set of possible causal effects. Since every causal effect is identifiable given a DAG and since we have computationally efficient strategies for enumeration, this is a sensible approach. The authors discuss the difficulty in the enumeration procedure for the latent variable case, but it would also be important to point out that as a consequence of Theorem 4 of Jaber et al. 2019 (ICML), the set identifiability is not possible in the same way in the hidden variable setting. That is, if the effect is not identifiable in a PAG, there will always be a MAG compatible with this PAG where this effect is not identifiable. Meaning that one would still need to bind some of these effects using the authors' results. Additionally, one such enumeration procedure already exists for the case of PAGs, namely: Malinsky and Spirtes, 2016: "Estimating Causal Effects with Ancestral Graph Markov Models". Indeed, the procedure described in this paper is computationally intense, but it could be used in combination with the authors' results to obtain some estimates.

- My other comment to the authors regards the assumption of faithfulness. The importance of the assumption of faithfulness is maintained throughout the paper, but I still don't fully understand why that is. It is clear to me that this assumption, or even the strong faithfulness assumption, is needed for causal discovery, but that is not the focus of this paper. Instead, this paper starts with knowing the PAG (somehow). It assumes this PAG is Markov and faithful, meaning that there exists a causal DAG on the observed and unobserved variables and that this PAG is obtained by marginalizing out the unobserved variables and removing specific edge directions (that are not common across the ME).

In the DAG case, if a DAG G is not faithful, but it satisfies the Markov assumption and all edge directions for the faithful edges are correct, then all identification results in G also hold for the true DAG G', which is a subgraph of G. This is somewhat discussed in Proposition 7 of Peters and Bühlmann (2015) - Structural intervention distance. Hence, knowing an unfaithful DAG in this sense is not an issue for identification, though it may lead to some less desirable results in terms of efficiency.

Returning to the PAG setting, it is clear to me that if the PAG you are considering is not faithful but all faithful edges are correctly oriented, that may lead to bounds on the causal effect that are not tight. But will it lead to incorrect results? The identifability results for the PAG do rely on the presence of specific edges for the visibility of others, so this setting is not exactly equivalent to the DAG one, but wouldn't the bounds you propose still hold in the faithful PAG? I would appreciate a bit of a discussion on this in the camera-ready version. However, it may be that this question is too complicated to answer in a short comment in the paper, so I would not put this as a condition of acceptance.

In summary, this paper represents a great first step toward useful estimation methods when only relying on conditional independence constraints!